# Provably Faster Algorithms for Bilevel Optimization via Without-Replacement Sampling

**Junyi Li and Heng Huang**
Department of Computer Science, Institute of Health Computing
University of Maryland College Park
College Park, MD, 20742
junyili.ai@gmail.com, henghuanghh@gmail.com

## Abstract

Bilevel Optimization has experienced significant advancements recently with the introduction of new efficient algorithms. Mirroring the success in single-level optimization, stochastic gradient-based algorithms are widely used in bilevel optimization. However, a common limitation in these algorithms is the presumption of independent sampling, which can lead to increased computational costs due to the complicated hyper-gradient formulation of bilevel problems. To address this challenge, we study the example-selection strategy for bilevel optimization in this work. More specifically, we introduce a without-replacement sampling based algorithm which achieves a faster convergence rate compared to its counterparts that rely on independent sampling. Beyond the standard bilevel optimization formulation, we extend our discussion to conditional bilevel optimization and also two special cases: minimax and compositional optimization. Finally, we validate our algorithms over both synthetic and real-world applications. Numerical results clearly showcase the superiority of our algorithms.

## 1 Introduction

Bilevel optimization [51, 47] has received a lot of interest recently due to its wide-ranging applicability in machine learning tasks, including hyper-parameter optimization [39], meta-learning [57] and personalized federated learning [46]. A bilevel optimization problem is a type of nested two-level problems as follows:

$$\min_{x \in \mathbb{R}^p} h(x) \coloneqq f(x, y_x) \text{ s.t. } y_x = \arg\min_{y \in \mathbb{R}^d} g(x, y), \tag{1}$$

which includes an outer problem $f(x, y)$ and a inner problem $g(x, y)$. The outer problem $f(x, y)$ relies on the solution $y_x$ of the inner problem $g(x, y)$. Eq. (1) can be solved through gradient descent: $x_{t+1} = x_t - \eta \nabla h(x_t)$, with $\eta$ be the stepsize and $\nabla h(x_t)$ be gradient at the state $x_t$. Specially, the gradient $\nabla h(x)$ [17] is a function of the inner problem minimizer $y_x$, first order derivatives $\nabla_x f(x, y)$, $\nabla_y f(x, y)$, $\nabla_y g(x, y)$ and second order derivatives $\nabla_{xy} g(x, y)$, $\nabla_{y^2} g(x, y)$. In particular, $y_x$ can be solved though gradient descent: $y_{t+1} = y_t - \gamma \nabla_y g(x, y_t)$ with $\gamma$ be the stepsize and $\nabla_y g(x_t, y_t)$ be gradient at the iterate $(x_t, y_t)$. In the standard setup, the outer and inner problems are defined over two datasets $\mathcal{D}_u = \{\xi_i, i \in [m]\}$ and $\mathcal{D}_l = \{\zeta_j, i \in [n]\}$, respectively:

$$f(x, y) = \frac{1}{m} \sum_{i=1}^{m} f(x, y; \xi_i), g(x, y) = \frac{1}{n} \sum_{j=1}^{n} g(x, y; \zeta_j),$$

Naturally, we sample from the datasets to estimate the first and second-order derivatives in the hyper-gradient formulation. And to guarantee convergence, previous approaches require the examples

38th Conference on Neural Information Processing Systems (NeurIPS 2024).

Table 1: **Comparisons of the Bilevel Opt. & Conditional Bilevel Opt. algorithms for finding an $\epsilon$-stationary point**. We include comparison with stochastic gradient descent type of methods and a more comprehensive comparison including other acceleration methods can be found in Table 2 of Appendix. An $\epsilon$-stationary point is defined as $\|\nabla h(x)\| \leq \epsilon$. $Gc(f, \epsilon)$ and $Gc(g, \epsilon)$ denote the number of gradient evaluations *w.r.t.* $f(x, y)$ and $g(x, y)$; $JV(g, \epsilon)$ denotes the number of Jacobian-vector products; $HV(g, \epsilon)$ is the number of Hessian-vector products. $m$ and $n$ are the number of data examples for the outer and inner problems, in particular, $n$ is the maximum number of inner problems for conditional bilevel optimization. Our methods have a dependence over example numbers $(max(m, n))^q$, $q$ is a value decided by without-replacement sampling strategy and can have value in $[0, 1]$ (A herding-based permutation [33] can let $q = 0$).

| Setting | Algorithm | $Gc(f, \epsilon)$ | $Gc(g, \epsilon)$ | $JV(g, \epsilon)$ | $HV(g, \epsilon)$ |
|---|---|---|---|---|---|
| B.O. | BSA [17] | $O(\epsilon^{-4})$ | $O(\epsilon^{-6})$ | $O(\epsilon^{-4})$ | $O(\epsilon^{-4})$ |
| | TTSA [22] | $O(\epsilon^{-5})$ | $O(\epsilon^{-5})$ | $O(\epsilon^{-5})$ | $O(\epsilon^{-5})$ |
| | StocBiO [25] | $O(\epsilon^{-4})$ | $O(\epsilon^{-4})$ | $O(\epsilon^{-4})$ | $O(\epsilon^{-4})$ |
| | SOBA [9] | $O(\epsilon^{-4})$ | $O(\epsilon^{-4})$ | $O(\epsilon^{-4})$ | $O(\epsilon^{-4})$ |
| | AID/ITD [25] | $O(max(m, n)\epsilon^{-2})$ | $O(max(m, n)\epsilon^{-2})$ | $O(max(m, n)\epsilon^{-2})$ | $O(max(m, n)\epsilon^{-2})$ |
| | **WiOR-BO(Ours)** | $O((max(m, n))^q\epsilon^{-3})$ | $O((max(m, n))^q\epsilon^{-3})$ | $O((max(m, n))^q\epsilon^{-3})$ | $O((max(m, n))^q\epsilon^{-3})$ |
| Cond. B.O. | DL-SGD [23] | $O(\epsilon^{-4})$ | $O(\epsilon^{-6})$ | $O(\epsilon^{-4})$ | $O(\epsilon^{-4})$ |
| | RT-MLMC [23] | $O(\epsilon^{-4})$ | $O(\epsilon^{-4})$ | $O(\epsilon^{-4})$ | $O(\epsilon^{-4})$ |
| | **WiOR-CBO (Ours)** | $O((max(m, n))^q\epsilon^{-3})$ | $O((max(m, n))^{2q}\epsilon^{-4})$ | $O(n(max(m, n))^q\epsilon^{-3})$ | $O((max(m, n))^{2q}\epsilon^{-4})$ |

be *mutually independent* [25] even at the same state $(x, y)$. For example, two different examples $\zeta_1$ and $\zeta_2$ are sampled to estimate $\nabla_y g(x, y)$ and $\nabla_{y^2} g(x, y)$, respectively. This leads to extra computational cost in practice. Indeed, two backward passes are required to evaluate $\nabla_x f(x, y)$ and $\nabla_y f(x, y)$, while five backward passes are needed to evaluate $\nabla_y g(x, y)$, $\nabla_{y^2} g(x, y)$ and $\nabla_{xy} g(x, y)$ for a given state of $(x, y)$. In contrast, one backward pass are needed to evaluate the former and two backward passes for the latter if not sampling independently for each property. This leads to notable computational cost difference for the current large-scale machine learning models [4]. Therefore, it is beneficial to explore other example-selection strategies beyond independent sampling. More specifically, we aim to answer the following question in this work: *Can we solve the bilevel optimization problem without using independent sampling?*

Although example-selection is under-explored for bilevel optimization, it has been well studied in the single level optimization literature [2, 42, 44]. For single level problems: $\min_{x \in \mathbb{R}^p} f(x) = \frac{1}{n} \sum_{i=1}^{n} f(x; \xi_i)$, we iteratively perform the stochastic gradient step $x_{t+1} = x_t - \eta \nabla f(x_t, \xi_t)$, where the sample gradient is used to estimate the full gradient and the example $\xi_t$ is sampled according to some order. More specifically, the sampling schemes are roughly divided into two categories: *with-replacement* sampling and *without-replacement* sampling. Traditional stochastic gradient descent typically employs with-replacement sampling to ensure an unbiased estimation of the full gradient in each iteration. In contrast, the without-replacement sampling has biased estimation at each iteration. Despite this, when averaging the gradients of consecutive examples, without-replacement sampling can approximate the full gradient more efficiently [34]. In fact, any permutation-based without-replacement sampling has that the gradient estimation error contracts at rate of $O(K^{-2})$ [34] where $K$ is the number of steps, while stochastic gradient descent only has a rate of $O(K^{-1})$. Inspired by the properties of without-replacement sampling demonstrated in the single-level optimization, we design the algorithm named WiOR-BO (and its variants) which performs without-replacement sampling for bilevel optimization. Naturally, WiOR-BO does not require the independent sampling as in previous methods. In fact, our algorithm enjoys favorable convergence rate. As shown in Table 1, WiOR-BO converges faster compared with their counterparts (*e.g.* StocBiO [25]) using independent sampling. Actually, WiOR-BO gets comparable convergence rate with methods using more complicated variance reduction techniques (*e.g.* MRBO [52]). In Table 2, we perform a more comprehensive comparison with other variance reduction techniques and in Section C of Appendix, we further compare WiOR-BO with these methods.

The **contributions** of our work can be summarized as:

1. We propose **WiOR-BO** which performs without-sampling to solve the bilevel optimization problem and converges to an $\epsilon$ stationary point with a rate of $O(\epsilon^{-3})$. This rate improves over the $O(\epsilon^{-4})$ rate of its counterparts (*e.g.* stocBiO [25]) using independent sampling.

2. We propose **WiOR-CBO** for the conditional bilevel optimization problems and show that our algorithm converges to an $\epsilon$ stationary point with a rate of $O(\epsilon^{-4})$. This rate improves over the $O(\epsilon^{-6})$ rate of its counterparts (*e.g.* DL-SGD [23]) using independent-sampling.

3. We customize our algorithms to the special cases of minimax and compositional optimization problems and demonstrate similar favorable convergence rates.

**Notations.** $\nabla$ ($\nabla_x$) denotes full gradient (partial gradient, over variable $x$), higher order derivatives follow similar rules. $[n]$ represents sequence of integers from 1 to $n$. $O(\cdot)$ is the big O notation, and we hide logarithmic terms.

## 2 Related Works

**Bilevel Optimization** has its roots tracing back to the 1960s, beginning with the regularization method proposed by [51], and then followed by many research works [14, 47, 43, 55]. In the field of machine learning, similar implicit differentiation techniques were used to solve Hyper-parameter Optimization [28, 6, 12]. Exact solutions for Bilevel Optimization solve the inner problem for each outer variable but are inefficient. More efficient algorithms solve the inner problem with a fixed number of steps, and use the 'back-propagation through time' technique to compute the hyper-gradient [13, 35, 16, 40, 45]. Recently, single loop algorithms [17, 22, 25, 26, 52, 9, 30, 24] are introduced to perform the outer and inner updates alternatively. Besides, other aspects of the bilevel optimization are also investigated: [23] studied a type of conditional bilevel optimization where the inner problem depends on the example of the outer problem; [53, 56] studied the Hessian-free bilevel optimization and proposed an algorithm with optimal convergence rate; [32] proposed an efficient single loop algorithm for the general non-convex bilevel optimization; [54, 31] studies the bilevel optimization in the federated learning setting.

**Sample-selection** in stochastic optimization has been extensively studied for the case of single level optimization problems [2, 3, 19, 38]. Beyond the with-replacement sampling adopted by SGD, various without-replacement strategies were also studied in the literature. The random-reshuffling strategy shuffles the data samples at the start of each training epoch, and its property was first studied in [42] via the non-commutative arithmetic-geometric mean conjecture and followed by [20, 19]. [44, 36] studied the shuffling-once strategy, where the data samples are only shuffled at the start of the training once; [8] investigated data echoing where one data sample is repeatedly used. Quasi-Monte Carlo sampling is also applied in stochastic gradient descent [5, 34]. Recently, [34, 33] proposed to use average gradient error to study the convergence of various example order strategies and then proposed a herding-based sampling strategy [50] to proactively minimize the average gradient error.

## 3 Without-Replacement Sampling in Bilevel Optimization

In this work, we focus on the following finite-sum bilevel optimization problem:

$$\min_{x \in \mathbb{R}^p} h(x) := f(x, y_x) = \frac{1}{m} \sum_{i=1}^{m} f(x, y_x; \xi_i), \text{ s.t. } y_x = \arg\min_{y \in \mathbb{R}^d} g(x, y) = \frac{1}{n} \sum_{j=1}^{n} g(x, y; \zeta_j) \quad (2)$$

where both the outer and the inner problems are defined over a finite dataset, and we denote them as $\mathcal{D}_u = \{\xi_i, i \in [m]\}$ and $\mathcal{D}_l = \{\zeta_j, j \in [n]\}$, respectively. Under mild assumptions, the hyper-gradient of $h(x)$ is:

$$\nabla h(x) = \nabla_x f(x, y_x) - \nabla_{xy} g(x, y_x) u_x, \ u_x = \nabla_{y^2} g(x, y_x)^{-1} \nabla_y f(x, y_x) \quad (3)$$

Our objective is to minimize $h(x)$ by exploiting Eq. (3) with examples from $\mathcal{D}_u$ and $\mathcal{D}_l$ at each step. More specifically, for a given example order $\pi$ denoting the following example-selection sequence as $\{\xi_t^\pi \in \mathcal{D}_u\}$ and $\{\zeta_t^\pi \in \mathcal{D}_l\}$ for $t \in [T]$, where $T$ denotes the length of the sequence and can be arbitrarily large. We then perform the following update steps iteratively for $t \in [T]$:

$$y_{t+1} = y_t - \gamma_t \nabla_y g(x_t, y_t; \zeta_t^\pi) \quad (4a)$$
$$u_{t+1} = u_t - \rho_t (\nabla_{y^2} g(x_t, y_t; \zeta_t^\pi) u_t - \nabla_y f(x_t, y_t; \xi_t^\pi)) \quad (4b)$$
$$x_{t+1} = x_t - \eta_t (\nabla_x f(x_t, y_t; \xi_t^\pi) - \nabla_{xy} g(x_t, y_t; \zeta_t^\pi) u_t) \quad (4c)$$

where Eq. (4) adopts the single-loop [30] update. More specifically, Eq. (4a) performs gradient descent to estimate the minimizer $y_x$; Eq. (4b) performs gradient descent to estimate $u_x$ defined in Eq. (3); and Eq. (4c) perform the gradient descent step given the estimation of $y_x$ and $u_x$.

---

**Algorithm 1** Without-Replacement Bilevel Optimization (WiOR-BO)

---

1: **Input:** Initial states $x_I^0$, $y_I^0$ and $u_I^0$; learning rates $\{\gamma_i^r, \rho_i^r, \eta_i^r\}, i \in [I], r \in [R], I := \mathrm{lcm}(m, n)$ denotes the least common multiple of $m$ and $n$.
2: **for** epochs $r = 1$ **to** $R$ **do**
3:     Randomly sample $I/m$ permutations of outer dataset and concatenate them to have $\{\xi_i^\pi, i \in [I]\}$, sample $I/n$ permutations of inner dataset and concatenate them to have $\{\zeta_i^\pi, i \in [I]\}$;
4:     Set $y_0^r = y_I^{r-1}$, $u_0^r = \mathcal{P}_\iota(u_I^{r-1})$ and $x_0^r = x_I^{r-1}$
5:     **for** $i = 0$ **to** $I - 1$ **do**
6:         $y_{i+1}^r = y_i^r - \gamma_i^r \nabla_y g(x_i^r, y_i^r; \zeta_i^\pi)$; $u_{i+1}^r = u_i^r - \rho_i^r(\nabla_{y^2} g(x_i^r, y_i^r; \zeta_i^\pi) u_i^r - \nabla_y f(x_i^r, y_i^r; \xi_i^\pi))$;
7:         $x_{i+1}^r = x_i^r - \eta_i^r(\nabla_x f(x_i^r, y_i^r; \xi_i^\pi) - \nabla_{xy} g(x_i^r, y_i^r; \zeta_i^\pi) u_i^r)$;
8:     **end for**
9: **end for**

---

Note that at each step $t$, we use a pair of examples $\{\xi_t^\pi, \zeta_t^\pi\}$ to estimate the involved properties and require only three backward passes, *i.e.* $\nabla f(x_t, y_t; \xi_t^\pi)$, $\nabla g(x_t, y_t; \zeta_t^\pi)$ and $\nabla(\nabla_y g(x_t, y_t; \zeta_t^\pi))$. In contrast, if we independently sample examples for each property, seven backward passes are needed. For large-scale machine learning models where back-propagation is expensive, our method can lead to significant computation saving in practice.

Equation (4) is independent of the specific order of samples. It can accommodate both with-replacement and without-replacement sampling strategies. In particular, we consider two most widely-used without-replacement sampling methods: *random-reshuffling* and *shuffle-once*. For random-reshuffling, we set $T = R \times lcm(m, n)$, where $lcm(m, n)$ denotes the least common multiple of $m$ and $n$ and $R$ is a constant. More specifically, we generate $R \times lcm(m, n)/m$ random permutations of examples in $\mathcal{D}_u$ then concatenate them to get $\{\xi_t^\pi \in \mathcal{D}_u\}$, and we follow similar procedure to generate $\{\zeta_t^\pi \in \mathcal{D}_u\}$. In algorithm 1, we show our **WiOR-BO** algorithm for solving Eq. (2) with random-reshuffling sampling. Note that in Line 4 of Algorithm 1, $\mathcal{P}_\iota(\cdot)$ is the projection operation to an $\iota$-size ball and we perform projection to make $u_t$ be bounded during updates. Shuffle-once is another widely used without-replacement sampling strategy, where a single permutation is generated and reused for both $\mathcal{D}_u$ and $\mathcal{D}_l$, and we can modify Line 3 of Algorithm 1 to incorporate shuffle-once. Finally, compared to independent sampling-based algorithm such as stocBiO [25], we require extra space to generate and store the orders of examples, but this cost is negligible compared to memory and computational cost of training large scale models.

### 3.1 Conditional Bilevel Optimization

In Eq. (2), we assume the outer dataset $\mathcal{D}_u$ and the inner dataset $\mathcal{D}_l$ are independent with each other. However, it is also common in practice that the inner problem not only relies on the outer variable $x$, but also the current outer example. This type of problems is known as conditional (contextual) bilevel optimization problems [23] in the literature and we consider the finite setting as follows:

$$\min_{x \in \mathbb{R}^p} h(x) := f(x, y_x) = \frac{1}{m}\sum_{i=1}^m f(x, y_x^{(\xi_i)}, \xi_i), \text{s.t. } y_x^{(\xi_i)} = \arg\min_{y \in \mathbb{R}^d} g^{(\xi_i)}(x, y) = \frac{1}{n_{\xi_i}}\sum_{j=1}^{n_{\xi_i}} g(x, y; \zeta_{\xi_i, j}),$$
(5)

Note that in Eq. (5), the outer problem is defined on a finite dataset $\mathcal{D}_u = \{\xi_i, i \in [m]\}$, and for each example $\xi_i \in \mathcal{D}_u$, we have a inner problem $g^{(\xi_i)}(x, y)$ defined on a dataset $\mathcal{D}_{l,\xi_i} = \{\zeta_{\xi_i, j}, j \in [n_{\xi_i}]\}$. The hyper-gradient of Eq. (5) is:

$$\nabla h(x) = \frac{1}{m}\sum_{i=1}^m \left(\nabla_x f(x, y_x^{\xi_i}; \xi_i) - \nabla_{xy} g^{(\xi_i)}(x, y_x^{\xi_i}) u_x^{\xi_i}\right), u_x^{\xi_i} = \nabla_{y^2} g^{(\xi_i)}(x, y_x)^{-1} \nabla_y f(x, y_x; \xi_i),$$
(6)

The key difference of Eq. (6) with Eq. (3) is that $y_x^{\xi_i}$ and $u_x^{\xi_i}$ are defined for each example $\xi_i \in \mathcal{D}_u$. To minimize $h(x)$ in Eq. (5), we assume a sample order $\pi$ denoting the following example selection sequence as $\{\xi_t^\pi \in \mathcal{D}_u, t \in [T]\}$, where $T$ denotes the example sequence length and can be arbitrarily

---

**Algorithm 2** Without-Replacement Conditional Bilevel Optimization (WiOR-CBO)

---

1: **Input:** Initial state $x^0$, learning rates $\{\eta_i^r\}, i \in [m], r \in [R]$.
2: **for** epochs $r = 1$ **to** $R$ **do**
3:     Randomly sample a permutation of outer dataset to have $\{\xi_i^\pi, i \in [m]\}$ and set $x_0^r = x_m^{r-1}$ or $x_0$ if $r = 1$.
4:     **for** $i = 0$ **to** $m - 1$ **do**
5:         Initial states $y^0$ and $u^0$, learning rates $\{\gamma_j^s, \rho_j^s\}, j \in [n_i], s \in [S]$.
6:         **for** $s = 1$ **to** $S$ **do**
7:             Sample a permutation of inner dataset to have $\{\zeta_{\xi_i,j}^\pi, j \in [n_i]\}$ , set $y_0^s = y_{n_i}^{s-1}$ and $u_0^s = \mathcal{P}_\iota(u_{n_i}^{s-1})$ or $y_0^s = y_0$ and $u_0^s = u_0$ if $s = 1$.
8:             **for** $j = 0$ **to** $n_i - 1$ **do**
9:                 $y_{j+1}^s = y_j^s - \gamma_j^s \nabla_y g(x_i^r, y_j^s; \zeta_{\xi_i,j}^\pi);$
10:                $u_{j+1}^s = u_j^s - \rho_j^s(\nabla_{y^2} g(x_i^r, y_j^s; \zeta_{\xi_i,j}^\pi) u_j^s - \nabla_y f(x_i^r, y_j^s; \xi_i^\pi));$
11:            **end for**
12:        **end for**
13:        $x_{i+1}^r = x_i^r - \eta_i^r(\nabla_x f(x_i^r, y_{n_i}^S; \xi_i^\pi) - \nabla_{xy} g^{\xi_i^\pi}(x_i^r, y_{n_i}^S) u_{n_i}^S);$
14:    **end for**
15: **end for**

---

large. We then perform the following update steps iteratively:

$$x_{t+1} = x_t - \eta_t(\nabla_x f(x_t, \hat{y}_t; \xi_t^\pi) - \nabla_{xy} g^{\xi_t^\pi}(x_t, \hat{y}_t)\hat{u}_t) \tag{7}$$

where $\hat{y}_t$ and $\hat{u}_t$ are estimations of $y_{x_t}^{\xi_t^\pi}$ $u_{x_t}^{\xi_t^\pi}$. We get them by performing the following rules $T_l$ steps over the sample order $\{\zeta_{\xi_t^\pi,t_l}^\pi \in \mathcal{D}_{l,\xi_t^\pi}, t_l \in [T_l]\}$:

$$y_{t_l+1} = y_{t_l} - \gamma_{t_l}\nabla_y g(x_t, y_{t_l}; \zeta_{\xi_t^\pi,t_l}^\pi), u_{t_l+1} = u_{t_l} - \rho_{t_l}(\nabla_{y^2} g(x_t, y_{t_l}, \zeta_{\xi_t^\pi,t_l}^\pi)u_{t_l} - \nabla_y f(x_t, y_{t_l}; \xi_t^\pi)) \tag{8}$$

where both the outer variable state $x_t$ and sample $\xi_t^\pi$ are fixed. We then set $\hat{y}_t = y_{T_l}$ and $\hat{u}_t = u_{T_l}$ in Eq. (7). Note that Eq. (7) and Eq. (8) perform a double-loop update rule [23]. Since the inner problem relies on the example of the outer problem, we compute $y_x^{\xi_i}$ and $u_x^{\xi_i}$ for each new state $x$ and sample $\xi_i$ instead of reusing them as in the single loop update of Eq. (4).

Note that an important feature of our algorithm is that we select samples based on an example order $\pi$ at each step instead of sampling independently. Similar to the unconditional bilevel optimization, various example orders can be incorporated. For random-reshuffling, we set $T = Rm$ with $R$ be a constant. Then we generate $R$ random permutations of examples in $\mathcal{D}_u$ and concatenate them to get the example order $\{\xi_t^\pi \in \mathcal{D}_u, t \in [T]\}$. Similarly, we concatenate $S$ random permutations of the inner dataset $\mathcal{D}_{l,\xi_t}$ to get $\{\zeta_{\xi_t^\pi,t_l}^\pi \in \mathcal{D}_{l,\xi_t^\pi}, t_l \in [T_l]\}$ with a sequence length $T_l = S \times n_{\xi_i}$. We summarize this in Algorithm 2 and denote it as **WiOR-CBO**. We slightly abuse the notation to replace $n_{\xi_i}$ with $n_i$ in the inner loop update for better clarity.

### 3.2 MiniMax and Compositional Optimization Problems

Our discussion of without-replacement sampling for bilevel optimization can be customized for two important nested optimization problems: *compositional optimization* and *minimax optimization* problems, as they can be viewed as a special type of bilevel optimization problem. Suppose we let the outer problem only rely on $y$, set the inner problem as $g(x, y) = \frac{1}{2}\|y - r(x)\|^2$. Consider the finite case of $f(y)$ and $r(x)$, we have the following compositional optimization problem:

$$\min_{x \in \mathbb{R}^p} h(x) := \frac{1}{m}\sum_{i=1}^m f(r(x); \xi_i) = \frac{1}{m}\sum_{i=1}^m f(\frac{1}{n}\sum_{j=1}^n r(x; \zeta_j); \xi_i) \tag{9}$$

Then we can derive the hyper-gradient as: $\nabla h(x) = \nabla_x r(x) u_x$, where $u_x = \nabla_y f(r(x))$, meanwhile, we have $\nabla_y g(x, y) = y - r(x)$. Then we can instantiate Algorithm 1 to get an algorithm for compositional optimization problem using without-replacement sampling (Algorithm 3 of Appendix). Similar to the SCGD algorithm [49], we track the exponential moving average of the inner state

$y$, besides we also track the exponential moving average of inner gradient (the update rule of $u$ in Algorithm 3).

Next, similar to the unconditional case, the conditional compositional optimization problem can be specialized from the conditional bilevel optimization problem Eq. (5) to have:

$$\min_{x \in \mathbb{R}^p} h(x) \coloneqq \frac{1}{m} \sum_{i=1}^{m} f(\frac{1}{n_i} \sum_{j=1}^{n_i} r(x; \zeta_{\xi_i,j}); \xi_i) \tag{10}$$

and we can instantiate Algorithm 2 in the special case of Eq. (10) to get Algorithm 5 (in the Appendix).

The minimax optimization can also be viewed as a special type of bilevel optimization. More specifically, if we have $g(x, y) = -f(x, y)$ in Eq. (2), then we get the following minimax problem:

$$\min_{x \in \mathbb{R}^p} h(x) \coloneqq \frac{1}{m} \sum_{i=1}^{m} \max_{y \in \mathbb{R}^d} f(x, y; \xi_i) \tag{11}$$

In the special case of Eq. (11), the hyper-gradient $\nabla h(x)$ in Eq. (3) degenerates to $\nabla h(x) = \nabla_x f(x, y_x)$ due to the fact that $\nabla_y f(x, y_x) = -\nabla_y g(x, y_x) = 0$ by the optimality condition. Meanwhile, we have $\nabla_y g(x, y) = -\nabla_y f(x, y)$. As a result, we can simplify Algorithm 1 to get an alternative update algorithm for minimax optimization using without-replacement sampling (Algorithm 4 in the Appendix). Note that due to the linearity of minimax problem *w.r.t.* examples, the conditional case for minimax optimization degenerates to the unconditional case. [11, 7] also considers the without-replacement sampling for the minimax problem, however, they focus on the strongly-convex-strongly-concave and two-sided Polyak-Łojasiewicz settings, in contrast, our WiOR-MiniMax can be applied to nonconvex-strongly-concave setting.

# 4 Theoretical Analysis

In this section, we provide theoretical analysis of our algorithms. We first state some assumptions:

**Assumption 4.1.** Function $f(x, y; \xi), \xi \in \mathcal{D}_u$, is possibly non-convex, $L$-smooth and has $C_f$-bounded gradient *w.r.t.* the variable $y$.

**Assumption 4.2.** Function $g(x, y; \zeta), \zeta \in \mathcal{D}_l$, is $\mu$-strongly convex *w.r.t* $y$ for any given $x$, and $L$-smooth. Furthermore, $\nabla_{xy} g(x, y; \zeta)$ and $\nabla_{y^2} g(x, y; \zeta)$ are Lipschitz continuous with constants $L_{xy}$ and $L_{y^2}$ respectively.

**Assumption 4.3.** For all examples $\xi_i \in D_u$ and states $(x, y) \in \mathbb{R}^{p+d}$, there exists an upper bound A such that the sample gradient errors satisfy $\|\nabla f(x, y; \xi_i) - \nabla f(x, y)\| \le A$.

**Assumption 4.4.** For all examples $\zeta_j \in D_l$ and states $(x, y) \in \mathbb{R}^{p+d}$, there exists an upper bound A such that the sample gradient errors satisfy $\|\nabla g(x, y; \zeta_i) - \nabla g(x, y)\| \le A$ and $\|\nabla^2 g(x, y; \zeta_i) - \nabla^2 g(x, y)\| \le A$.

As stated in The assumption 4.1 and 4.2, we consider the *non-convex-strongly-convex* bilevel optimization problems, this class of problems is widely studied in bilevel optimization literature [17, 25].Assumption 4.3 and 4.4 guarantee that the example gradients are good estimation of full gradient and is widely used in the analysis of single-level finite-sum optimization problems [36]. Note that as the hyper-gradient (Eq. 3) involves second order derivatives, Assumption 4.4 also requires that the second order example gradients have bounded bias.

We next make the following assumptions about the example selection sequence. We assume $\{\xi_t^\pi \in \mathcal{D}_u, t \in [T]\}$ and $\{\zeta_t^\pi \in \mathcal{D}_l, t \in [T]\}$ satisfy:

**Assumption 4.5.** Given sequences of samples: $\{\xi_t^\pi \in \mathcal{D}_u, t \in [T]\}$ and $\{\zeta_t^\pi \in \mathcal{D}_l, t \in [T]\}$, we assume there exists some constants $\alpha, C$ such that for any step $t$ and any interval $k > 0$, we have:

$$\left\| \frac{1}{k} \sum_{\tau=t}^{t+k-1} \nabla f(x_t, y_t; \xi_\tau^\pi) - \nabla f(x_t, y_t) \right\|^2 \le k^{-\alpha} C^2$$

and similar bounds hold for $\nabla g$ and $\nabla^2 g$ (See Appendix B.1 for the full version of this assumption.)

Assumption 4.5 measures how well the sample gradients approximate the full gradient over an interval on average. Similar assumptions are used to study the sample order in single level optimization problems [34]. If samples are selected independently at each step, we get $\alpha = 1$ and $C = A$ based on Assumption 4.3 and 4.4. For any permutation-based order (*e.g.* random-reshuffling and shuffle-once), we can show the assumption holds with $\alpha = 2$ and $C = \max(m, n) \times A$. Note that the $C$ depends on the number of samples $m$ ($n$), this dependence can be reduced to $(\max(m, n))^{0.5}$ for sufficiently small learning rates [34, 37]. More recently, [33] proposed a herding-based algorithm which explicitly optimizes the gradient error in Assumption 4.5 and shows that we can reach $\alpha = 2$ and $C = O(A)$.

**Bilevel Optimization**. We are ready to show the convergence of Algorithm 1. Firstly, we denote the following potential function $\mathcal{G}_t := \mathcal{G}_t = h(x_t) + \phi_y \|y_t - y_{x_t}\|^2 + \phi_u \|u_t - u_{x_t}\|^2$, where $\phi_y$ and $\phi_u$ are constants that relates to the smoothness parameters of $h(x)$. Then we have:

**Theorem 4.6.** *Suppose Assumptions 4.1-4.4 are satisfied, and Assumption 4.5 is satisfied with some $\alpha$ and $C$ for the example order we use in Algorithm 1. We choose learning rates $\eta_t = \eta$, $\gamma_t = c_1\eta$, $\rho_t = c_2\eta$ and denote $T = R \times I$ be the total number of steps. Then for any pair of values $(E, k)$ which has $T = E \times k$ and $\eta \leq \frac{1}{k\tilde{L}_0}$, we have:*

$$\frac{1}{E} \sum_{e=0}^{E-1} \|\nabla h(x_{ke})\|^2 \leq \frac{2\Delta}{kE\eta} + 2\tilde{L}^2 k^{-\alpha} C^2$$

*where $c_1$, $c_2$, $\tilde{L}_0$, $\tilde{L}$ are constants related to the smoothness parameters of $h(x)$, $\Delta$ is the initial sub-optimality, $R$ and $I$ are defined in Algorithm 1.*

To reach an $\epsilon$ stationary point, we choose $\eta = \frac{1}{k\tilde{L}_0}$, and for $(E, k)$, we choose: $E = \frac{4\tilde{L}_0\Delta}{\epsilon^2}$ and $k = \left(\frac{4\tilde{L}^2 C^2}{\epsilon^2}\right)^{1/\alpha}$, which means we need to choose the number of epochs $R = \frac{T}{I} = \frac{4\tilde{L}_0(4\tilde{L}^2 C^2)^{1/\alpha}\Delta}{I\epsilon^{2+2/\alpha}}$. Specially, if we choose examples independently at each step, we have $C = A$ and $\alpha = 1$, then we need $T = O(\epsilon^{-4})$ steps to reach an $\epsilon$-stationary point. This recovers the rate of stocBiO algorithm [25] and SOBA [9]. Next, if we use some permutation-based without-replacement sampling strategy witch has $C = (\max(m, n))^q \times A$ ($q \in [0, 1]$) and $\alpha = 2$, then we need $T = O((\max(m, n))^q \epsilon^{-3})$. In particular, by adopting a herding-based permutation as in [33], we get $q = 0$, and our WiOR-BO strictly outperforms the independent sampling-based stocBiO and SOBA algorithms.

**Conditional Bilevel Optimization**. Next, we study the convergence rate of Algorithm 2. Besides Assumptions 4.3 and 4.4, we need the bias of sample hyper-gradient be bounded too:

**Assumption 4.7.** *For all examples $\xi_i \in D_u$ and states $x \in \mathbb{R}^p$, there exists an upper bound A such that the sample gradient errors satisfy $\|\nabla h(x; \xi_i) - \nabla h(x)\| \leq A$.*

We also augment Assumption 4.5 to include the case of $\nabla h(x)$ (Seem Appendix B.1 for more details.). Then we are ready to show the convergence rate, we use the potential function $\mathcal{G}_t = h(x_t)$ and have:

**Theorem 4.8.** *Suppose Assumptions 4.1-4.4 and 4.7 are satisfied, and Assumption 4.5 is satisfied with some $\alpha$ and $C$ for the example order we use in Algorithm 2. We choose learning rates $\eta_t = \eta = \frac{1}{8k\tilde{L}}$, $\gamma_t = \gamma = \frac{1}{256kL\kappa}$, $\rho_t = \rho = \frac{1}{512kL\kappa}$ and denote $T = R \times m$ be the total number of outer steps and $T_l = S \times \max(n_i)$ be the maximum inner steps. Then for any pair of values $(E, k)$ which has $T = E \times k$ and $T_l = E_l \times k$, we have:*

$$\frac{1}{E} \sum_{e=0}^{E-1} \|\nabla h(x_{ke})\|^2 \leq \frac{8\Delta_h}{kE\eta} + C_u(1 - \frac{k\mu\rho}{2})^{E_l}\Delta_u + C_y(1 - \frac{k\mu\gamma}{2})^{E_l}\Delta_y + (C')^2 C^2 k^{-\alpha}$$

*where $C_u$, $C_y$, $C'$, $\tilde{L}$ are constants related to the smoothness parameters of $h(x)$, $\Delta_h$, $\Delta_u$, $\Delta_y$ are the initial sub-optimality, $R$ and $S$ are defined in Algorithm 2.*

To reach an $\epsilon$ stationary point, we choose: $E = \frac{128\tilde{L}\Delta}{\epsilon^2}$, and , $k = \left(\frac{2(C')^2 C^2}{\epsilon^2}\right)^{1/\alpha}$ and $E_l = O(log(\epsilon^{-1}))$. Then the sample complexity of Algorithm 2 is $EE_l k^2 = O(C^{4/\alpha}\epsilon^{-(2+4/\alpha)})$, where we omit the logarithmic term $E_l$. Specially, if we choose examples independently at each step, then we have sample complexity $O(\epsilon^{-6})$ steps for an $\epsilon$-stationary point. This recovers the rate of DL-SGD algorithm [23]. Next, if we use some permutation-based without-replacement sampling strategy,

then we have sample complexity $O((\max(m,n))^{2q}\epsilon^{-4})$, which can match the state-of-art algorithm RT-MLMC [23] if we choose an example order with $q = 0$ [33].

**MiniMax and Compositional Optimization**. As special cases of Bilevel Optimization, we can also show that Algorithm 3, Algorithm 4 and Algorithm 5 converge under similar rate. Please see Corollary B.21 and Corollary B.20 for more details.

## 5  Applications and Numerical Experiments

In this section, we verify the effectiveness of the proposed algorithm through a synthetic invariant risk minimization task and two real world bilevel tasks: Hyper-Data Cleaning and Hyper-representation Learning. The code is written in Pytorch. Our experiments were conducted on servers equipped with 8 NVIDIA A5000 GPUs. Experiments are averaged over five independent runs, and the standard deviation is indicated by the shaded area in the plots.

### 5.1  Invariant Risk-Minimization

In this section, we consider a synthetic invariant risk minimization task [21]. More specifically, we perform logistic regression over a set of examples $\{(c_i, b_i), i \in [M]\}$ with $c$ be the input and $b$ be the target. However, we does not have access to $c$ but a set of noisy observations $\{c_{j,i}, j \in n_i\}$ for each $c_i$. To learn the coefficient $x$ between $c$ and $b$, we optimize the following objective:

$$\min_{x \in \mathbb{R}^p} \frac{1}{m} \sum_{i=1}^{m} \log(1 + \exp(-b_i y_x^{(i)})),$$

$$\text{s.t. } y_x^{(i)} = \arg\min_{y \in \mathbb{R}^d} \frac{1}{n_i} \sum_{j=1}^{n_i} \frac{1}{2}\|y - c_{j,i}x\|^2$$

Figure 1: Comparison of different sampling strategies for the Invariant Risk-Minimization task.

This is a conditional bilevel optimization problem and we can solve it using our WiOR-CBO. We generate synthetic data to test the effects of different sample order (sample generation process is described in Appendix A). More specifically, we compare independent-sampling (WiR-CBO), shuffle-once (WiOR-CBO (SO)) and random-reshuffling (WiOR-CBO (RR)). As shown in Figure 1, the two without-replacement sampling methods outperforms the independent sampling one.

### 5.2  Hyper-Data Cleaning

In the Hyper-Data Cleaning task, we are given noisy training dataset whose labels are corrupted by noise, meanwhile, we have a validation set whose labels are not corrupted. Our target is then using the clean validation samples to identify corrupted samples in the training set. More specifically, we learn optimal weights for training samples such that a model learned over the weighted training set performs well on the validation set. We can formulate this task as a bilevel optimization problem (Eq. (2)). A mathematical formulation of the task is included in Appendix A.

**Dataset and Baselines.** We construct datasets based on MNIST [29]. For the training set, we randomly sample 40000 images from the original training dataset and then randomly perturb a fraction of labels of samples. For the validation set, we randomly select 5000 clean images from the original training dataset. In our experiments, we test our WiOR-BO algorithm (Algorithm 1), with the shuffle-once (WiOR-BO-SO) and random-reshuffling (WiOR-BO-RR) sampling; additionally, we also consider the following non-adaptive bilevel algorithms as baselines: reverse [16], stocBiO [25], BSA [17], AID-CG [18], MRBO [52], VRBO [52] and WiR-BO (the variant of our WiOR-BO using independent sampling, which is similar to the single loop algorithms such as SOBA [9], AmIGO [1] and FLSA [30]). We perform grid search for hyper-parameters and report the best results.

We summarize the results in Figure 2. In the figure, we compare the validation loss and F1 score *w.r.t.* both Hyper-iterations and running time. As shown in the figure, the two variants of our algorithm WiOR-BO-SO and WiOR-BO-RR get similar performance and they both outperform other baselines. In particular, note that WiR-BO differs with WiOR-BO-SO (RR) only over the example order, but the

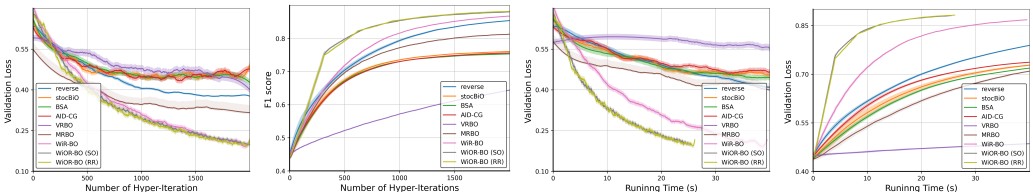

Figure 2: **Comparison of different algorithms for the Hyper-data Cleaning task.** The top two plots show validation error/F1 score vs Number of Hyper-Iterations and the bottom two plots show validation error/F1 score vs Running Time. The fraction of the noisy samples is 0.6.

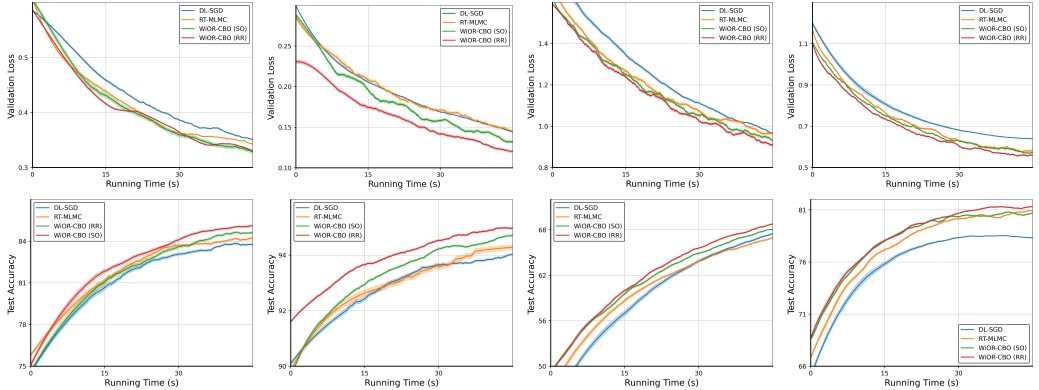

Figure 3: **Comparison of different algorithms for the Hyper-Representation Task over the Omniglot Dataset.** From Left to Right: 5-way-1-shot, 5-way-5-shot, 20-way-1-shot, 20-way-5-shot.

later converges much faster in terms of running time, this is partially due to less umber of backward passes needed by without-replacement sampling per iteration.

### 5.3 Hyper-Representation Learning

In this section, we consider Hyper-representation learning task. In this task, we learn a hyper-representation of the data such that a linear classifier can be learned quickly with a small number of samples. We consider the Omniglot [27] and MiniImageNet [41] data sets. This task can be viewed as a conditional bilevel optimization problem (Eq. (2)) and a mathematical formulation of the task is included in Appendix A.

**Dataset and Baselines.** The details of the datasets are included in Appendix A and we consider N-way-K-shot classification task following [48]. In our experiments, we test our WiOR-CBO (Algorithm 2), using the shuffle-once (WiOR-BO-SO) and random-reshuffling (WiOR-BO-RR) sampling. Besides, we compare with the following baselines: DL-SGD [23] and RT-MLMC [23]. Note that our WiOR-CBO can also use independent sampling, and it has similar performance as DL-SGD. We perform grid search of hyper-parameters for each method and report the best results.

We summarize the experimental results for the Omniglot dataset in Figure 3 and we defer the results for MiniImageNet to the Appendix A. As shown in the figure, our WiOR-CBO SO/RR outperforms DL-SGD and is comparable to the state-of-the-art algorithm RT-MLMC.

## 6 Conclusion

In this work, we investigated example-selection of bilevel optimization. Beyond the classical independent sampling, we assumed an example-order based on without-replacement sampling, such as random-reshuffling and shuffle-once. We proposed the WiOR-BO algorithm for bilevel optimization and show the algorithm converges to an $\epsilon$-stationary point with rate $O(\epsilon^{-3})$. After that, we also discussed the conditional bilevel optimization problems and introduced an algorithm with convergence rate of $O(\epsilon^{-4})$. As special cases of bilevel optimization, we studied the minimax and

compositional optimization problems. Finally, we validated the efficacy of our algorithms through one synthetic and two real-world tasks; the numerical results show the superiority of our algorithms.

## Acknowledgments

This work was partially supported by NSF IIS 2347592, 2347604, 2348159, 2348169, DBI 2405416, CCF 2348306, CNS 2347617.

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

# A    More Details for Numerical Experiments

In this section, we introduce more details of the experiments.

## A.1    Invariant Risk-Minimization

The data generation process is as follows: we first randomly sample a ground truth $x^*$, then we randomly generate $m = 1000$ input variables $c$, and calculate the ground truth label $b = cx$, then for each $c_i$, we generate a set of $n = 100$ noisy observations by adding Guassian noise of scale 0.1. We also add a $L_2$ regularization term of strength 0.1 for the outer problem. During trainging, we use both the inner and outer learing rates of 0.001.

## A.2    Hyper-Data Cleaning

The formulation of the problem is as follows:

$$\min_{x \in \mathbb{R}^p} h(x) := f(x, y_x) = \frac{1}{N^{(val)}} \sum_{n=1}^{N^{(val)}} \Theta(y_x; \xi_n^{val})$$

$$\text{s.t. } y_x = \arg\min_{y \in \mathbb{R}^d} g(x, y) = \sum_{n=1}^{N^{(tr)}} x_n \Theta(y; \xi_n^{tr})$$

In the above formulation, we have a pair of (noisy) training set $\{\xi_n^{tr}\}_{n=1}^{N^{(tr)}}$ and validation set $\{\xi_n^{val}\}_{n=1}^{N^{(val)}}$, and $x_n, n \in [N^{(tr)}]$ are weights for training samples, $y$ is the parameter of a model, and we denote the model by $\Theta$. Note that $y_x$ is the model learned over the weighted training set. We fit a model with 3 fully connected layers for the MNIST dataset. We also use $L_2$ regularization with coefficient $10^{-3}$ to satisfy the strong convexity condition. In the Experiments, we choose inner learning rate $(\gamma, \rho)$ as 0.1 and outer learning rate $\eta$ 1000.

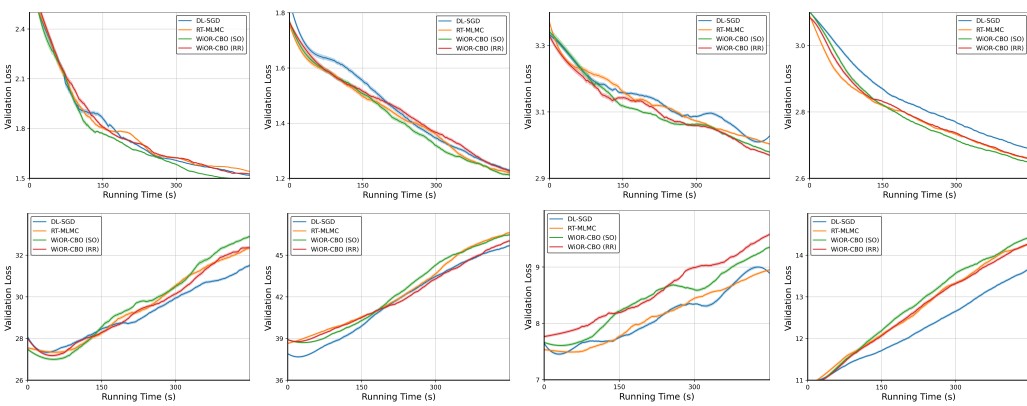

Figure 4: Comparison of different algorithms for the Hyper-Representation Task over the MiniImageNet Dataset. From Left to Right: 5-way-1-shot, 5-way-5-shot, 20-way-1-shot, 20-way-5-shot.

## A.3    Hyper-Representation Learning

$$\min_{x \in \mathbb{R}^p} h(x) := f(x, y_x) = \frac{1}{N} \sum_{n=1}^{N} \Big( \frac{1}{N_n^{val}} \sum_{i=1}^{N_n^{val}} \Theta(x, y_x^{(\mathcal{T}_n)}; \xi_i^{val}) \Big)$$

$$\text{s.t. } y_x^{(\mathcal{T}_n)} = \arg\min_{y \in \mathbb{R}^d} g^{(\mathcal{T}_n)}(x, y) = \frac{1}{N_n^{tr}} \sum_{i=1}^{N_n^{tr}} \Theta(x, y; \xi_i^{tr})$$

In the above formulation, we have $N$ tasks and each task $\mathcal{T}_n$ is defined by a pair of training set $\{\xi_i^{tr}\}_{i=1}^{N_n^{tr}}$ and validation set $\{\xi_i^{val}\}_{i=1}^{N_n^{val}}$. $\Theta$ defines the model, $x$ is the parameter of the backbone

model and $y$ is the parameter of the linear classifier. In summary, the lower level problem is to learn the optimal linear classifier $y$ given the backbone $x$, and the upper level problem is to learn the optimal backbone parameter $x$.

The Omniglot dataset includes 1623 characters from 50 different alphabets and each character consists of 20 samples. We follow the experimental protocols of [48] to divide the alphabets to train/validation/test with 33/5/12, respectively. We perform $N$-way-$K$-shot classification, more specifically, for each task, we randomly sample $N$ characters and for each character, we sample $K$ samples for training and 15 samples for validation. We augment the characters by performing rotation operations (multipliers of 90 degrees). We use a 4-layer convolutional neural network where each convolutional layer has 64 filters of 3×3 [15]. For the MiniImageNet, it has 64 training classes and 16 validation classes. Similar to Omniglot, we also perform the $N$-way-$K$-shot classification. We use a 4-layer convolutional neural network where each convolutional layer has 64 filters of 3×3 [15] for experiments. For the experiments, we use inner learning rates 0.4 and outer learning rates 0.1 for Omniglot related experiments and inner learning rates 0.01 and outer learning rates 0.05 for MiniImageNet-related experiments. We perform 4 inner gradient descent steps and set $K_m ax = 6$ for the RT-MLMC method. The experimental results for the MiniImageNet dataset is shown in Figure 4.

# B  Proof for Theorems

In this section, we provide proof for the convergence results in the main text. First, we restate all assumptions needed in our proof below:

**Assumption B.1** (Assumption 4.1). The function $f(x,y)$ is possibly non-convex and uniformly $L$-smooth, *i.e.* for any $x_1$, $x_2 \in \mathcal{X}$, $y_1, y_2 \in \mathbb{R}^d$, any $\xi \in \mathcal{D}_u$. Denote $z_1 = (x_1, y_1)$, $z_2 = (x_2, y_2)$, then we have:

$$f(z_1; \xi) \leq f(z_2; \xi) + \langle \nabla f(z_2, \xi), z_1 - z_2 \rangle + \frac{L}{2}||z_1 - z_2||^2.$$

or equivalently: $||\nabla f(z_1; \xi) - \nabla f(z_2; \xi)|| \leq L||z_1 - z_2||$. We also assume for any $x \in \mathcal{X}$ and any $y \in \mathbb{R}^d$, and we denote $z = (x, y)$, then we have $||\nabla_y f(z)|| \leq C_f$.

**Assumption B.2** (Assumption 4.2). Function $g(x,y)$ is uniformly $\mu$-strongly convex *w.r.t* $y$ for any given $x$ and uniformly $L$-smooth, *i.e.* for any $y_1, y_2 \in \mathbb{R}^d$ and $\zeta \in \mathcal{D}_l$, we have:

$$g(x, y_1; \zeta) \geq g(x, y_2; \zeta) + \langle \nabla_y g(x, y_2; \zeta), y_2 - y_1 \rangle + \frac{\mu}{2}||y_2 - y_1||^2.$$

and we have:

$$g(z_1; \zeta) \leq g(z_2; \zeta) + \langle \nabla g(z_2; \zeta), z_1 - z_2 \rangle + \frac{L}{2}||z_1 - z_2||^2.$$

equivalently: $||\nabla g(z_1; \zeta) - \nabla g(z_2; \zeta)|| \leq L||z_1 - z_2||$. Furthermore, we have $\nabla_{xy} g(x, y)$ and $\nabla_{y^2} g(x, y)$ are Lipschitz continuous with constant $L_{xy}$ and $L_{y^2}$ respectively, *i.e.* we have: $||\nabla_{xy} g(z_1; \zeta) - \nabla_{xy} g(z_2; \zeta)|| \leq L_{xy}||z_1 - z_2||$ and $||\nabla_{y^2} g(z_1; \zeta) - \nabla_{y^2} g(z_2; \zeta)|| \leq L_{y^2}||z_1 - z_2||$.

**Assumption B.3** (Assumption 4.3). For all examples $\xi_i \in D_u$ and states $(x, y) \in \mathbb{R}^{p+d}$, there exists an outer bound A such that the sample gradient errors satisfy $||\nabla f(x, y; \xi_i) - \nabla f(x, y)|| \leq A$.

**Assumption B.4** (Assumption 4.4). For all examples $\zeta_j \in D_l$ and states $(x, y) \in \mathbb{R}^{p+d}$, there exists an outer bound A such that the sample gradient errors satisfy $||\nabla g(x, y; \zeta_i) - \nabla g(x, y)|| \leq A$ and $||\nabla^2 g(x, y; \zeta_i) - \nabla^2 g(x, y)|| \leq A$.

**Assumption B.5** (Assumption 4.7). For all examples $\xi_i \in D_u$ and states $x \in \mathbb{R}^p$, there exists an upper bound A such that the sample gradient errors satisfy $||\nabla h(x; \xi_i) - \nabla h(x)|| \leq A$.

**Assumption B.6.** Given two sequences $\{\xi_t^\pi \in \mathcal{D}_u, t \in [T]\}$ and $\{\zeta_t^\pi \in \mathcal{D}_l, t \in [T]\}$, we assume there exists some constants $\alpha$, $C$ such that for any step $t$ and any $k > 0$, we have: $\left\|\frac{1}{k}\sum_{\tau=t}^{t+k-1}\nabla f(x_t, y_t; \xi_\tau^\pi) - \nabla f(x_t, y_t)\right\|^2 \leq k^{-\alpha}C^2$, $\left\|\frac{1}{k}\sum_{\tau=t}^{t+k-1}\nabla g(x_t, y_t; \zeta_\tau^\pi) - \nabla g(x_t, y_t)\right\|^2 \leq k^{-\alpha}C^2$, $\left\|\frac{1}{k}\sum_{\tau=t}^{t+k-1}\nabla^2 g(x_t, y_t; \zeta_\tau^\pi) - \nabla^2 g(x_t, y_t)\right\|^2 \leq k^{-\alpha}C^2$, $\left\|\frac{1}{k}\sum_{\tau=t}^{t+k-1}\nabla h(x_t; \xi_\tau^\pi) - \nabla h(x_t)\right\|^2 \leq k^{-\alpha}C^2$ (Only for the conditional bilevel optimization case).

---

**Algorithm 3** Without-Replacement Compositional Optimization (WiOR-Comp)

---

1: **Input:** Initial states $x_I^0$, $y_I^0$ and $u_I^0$; learning rates $\{\gamma_i^r, \rho_i^r, \eta_i^r\}, i \in [I], r \in [R], I = \text{lcm}(m, n)$.
2: **for** epochs $r = 1$ **to** $R$ **do**
3:      Generate sample order sequence $\{\xi_i^\pi\}$ and $\{\zeta_i^\pi\}$ for $i \in [I]$ as Line 3 of Algorithm 1;
4:      Set $y_0^r = y_I^{r-1}$, $u_0^r = \mathcal{P}_\iota(u_I^{r-1})$ and $x_0^r = x_I^{r-1}$
5:      **for** $i = 0$ **to** $I - 1$ **do**
6:          $y_{i+1}^r = (1 - \gamma_i^r)y_i^r + \gamma_i^r r(x_i^r, \zeta_i^\pi);\ u_{i+1}^r = (1 - \rho_i^r)u_i^r + \rho_i^r \nabla_y f(y_i^r; \xi_i^\pi);$
7:          $x_{i+1}^r = x_i^r - \eta_i^r \nabla r(x_i^r; \zeta_i^\pi)u_i^r;$
8:      **end for**
9: **end for**

---

---

**Algorithm 4** Without-Replacement MiniMax Opt. (WiOR-MiniMax)

---

1: **Input:** Initial states $x_m^0$, $y_m^0$; learning rates $\{\gamma_i^r, \eta_i^r\}, i \in [m], r \in [R]$.
2: **for** epochs $r = 1$ **to** $R$ **do**
3:      sample a permutation of outer dataset to have $\{\xi_i^\pi, i \in [m]\}$;
4:      Set $y_0^r = y_m^{r-1}$ and $x_0^r = x_m^{r-1}$
5:      **for** $i = 0$ **to** $m - 1$ **do**
6:          $y_{i+1}^r = y_i^r + \gamma_i^r \nabla_y f(x_i^r, y_i^r; \xi_i^\pi);$
7:          $x_{i+1}^r = x_i^r - \eta_i^r \nabla_x f(x_i^r, y_i^r; \xi_i^\pi);$
8:      **end for**
9: **end for**

---

### B.1 Proof for Bilevel Optimization Problems

Suppose we define:

$$\Phi(x, y) = \nabla_x f(x, y) - \nabla_{xy} g(x, y) \times [\nabla_{y^2} g(x, y)]^{-1} \nabla_y f(x, y)$$

Then we have $\nabla h(x) = \Phi(x, y_x)$. Furthermore, we have the following proposition:

**Proposition B.7.** *Suppose Assumptions 4.1 and 4.2 hold, the following statements hold:*

    *a) $y_x$ is Lipschitz continuous in $x$ with constant $\rho = \kappa$, where $\kappa = \frac{L}{\mu}$ is the condition number of $g(x, y)$.*

    *b) $\|\Phi(x_1; y_1) - \Phi(x_2; y_2)\|^2 \leq \hat{L}^2(\|x_1 - x_2\|^2 + \|y_1 - y_2\|^2)$, where $\hat{L} = O(\kappa^2)$.*

    *c) $h(x)$ is smooth in $x$ with constant $\bar{L}$ i.e., for any given $x_1, x_2 \in X$, we have $\|\nabla h(x_2) - \nabla h(x_1)\| \leq \bar{L}\|x_2 - x_1\|$ where $\bar{L} = O(\kappa^3)$.*

This is a standard results in bilevel optimization and we omit the proof here.

The main technique we use to analysis the convergence of Algorithm 1 is *aggregated analysis*, in other words, we perform analysis over a block of length $k$ instead of the more common per step analysis in stochastic bilevel optimization. Note that this technique is essential in analyzing without-replacement sampling for single-level optimization problems, and we extend them to the bilevel setting. More specifically, suppose the total number of training steps are $T := R \times I$, where $R$ and $I$ are denoted in Algorithm 1, then we consider a block of training steps $[t, t + k]$ for $t \in [T - k]$. Furthermore, we denote: $\bar{\eta}_t = \sum_{\tau=t}^{t+k-1} \eta_\tau$, and $\bar{\nu}_t = \frac{1}{\bar{\eta}_t} \sum_{\tau=t}^{t+k-1} \eta_\tau \nu_\tau$; $\bar{\gamma}_t = \sum_{\tau=t}^{t+k-1} \gamma_\tau$ and $\bar{w}_t = \frac{1}{\bar{\gamma}_t} \sum_{\tau=t}^{t+k-1} \gamma_\tau w_\tau$; $\bar{\rho}_t = \sum_{\tau=t}^{t+k-1} \rho_\tau$, and $\bar{q}_t = \frac{1}{\bar{\rho}_t} \sum_{\tau=t}^{t+k-1} \rho_\tau q_\tau$, where $\nu_t = \nabla_x f(x_t, y_t; \xi_t) - \nabla_{xy} g(x_t, y_t; ; \zeta_t)u_t$, $w_t = \nabla_y g(x_t, y_t; \zeta_t)$ and $q_t = \nabla_{y^2} g(x_t, y_t; \zeta_t)u_t - \nabla_y f(x_t, y_t; \xi_t)$. In particular, we denote $r_x(u)$ as:

$$r_x(u) = \frac{1}{2} u^T \nabla_{y^2} g(x, y_x)u - \nabla_y f(x, y_x)^T u,$$

by assumption it is $L$ smooth and $\mu$ strongly convex. It is straightforward to see that $u_x = \arg\min_{u \in \mathbb{R}^d} r_x(u)$, where $u_x$ is defined in Eq. (3). Finally, in the proof, we consider the general case of different orders for all involved example derivatives. In other words, we have $\{\xi_{t,1}\}, \{\xi_{t,2}\}$,

---

**Algorithm 5** Without-Replacement Conditional Compositional Optimization (WiOR-CComp)

---

1: **Input:** Initial state $x^0$, learning rates $\{\eta_i^r\}, i \in [m], r \in [R]$.
2: **for** epochs $r = 1$ **to** $R$ **do**
3:      Randomly sample a permutation of outer dataset to have $\{\xi_i^\pi, i \in [m]\}$ and set $x_0^r = x_m^{r-1}$ or $x_0$ if $r = 1$.
4:      **for** $i = 0$ **to** $m - 1$ **do**
5:          Initial states $y^0$ and $u^0$, learning rates $\{\gamma_j^s, \rho_j^s\}, j \in [n_i], s \in [S]$.
6:          **for** $s = 1$ **to** $S$ **do**
7:              Sample a permutation of inner dataset to have $\{\zeta_{\xi_i,j}^\pi, i \in [n_i]\}$, set $y_0^s = y_{n_i}^{s-1}$ and $u_0^s = \mathcal{P}_\epsilon(u_{n_i}^{s-1})$ or $y_0^s = y_0$ and $u_0^s = u_0$ if $s = 1$.
8:              **for** $j = 0$ **to** $n_i - 1$ **do**
9:                  $y_{j+1}^s = (1 - \gamma_j^s)y_j^s + \gamma_j^s r(x_i^r; \zeta_{\xi_i,j}^\pi)$;
10:                $u_{j+1}^s = (1 - \rho_j^s)u_j^s + \rho_j^s \nabla f(y_j^s; \xi_i^\pi)$;
11:              **end for**
12:          **end for**
13:          $x_{i+1}^r = x_i^r - \eta_i^r \nabla r^{\xi_i^\pi}(x_i^r)u_{n_i}^S$;
14:      **end for**
15: **end for**

---

$\{\zeta_{t,1}\}$, $\{\zeta_{t,2}\}$ and $\{\zeta_{t,3}\}$ for $\nabla_y f(x, y)$, $\nabla_x f(x, y)$, $\nabla_y g(x, y)$, $\nabla_{y^2} g(x, y)$, $\nabla_{xy} g(x, y)$ respectively. The two orders used by Algorithm 1 is a special case of the proof and is practically appealing as they can reuse computation as we discussed in the introduction.

**Lemma B.8.** *For $\tau > t \geq 0$, the bias of $\| \sum_{\bar\tau=t}^{\tau-1} \eta_{\bar\tau}(\nu_{\bar\tau} - \nabla h(x_t))\|^2$, $\| \sum_{\bar\tau=t}^{\tau-1} \rho_{\bar\tau}(\nabla r_{x_t}(u_t) - q_{\bar\tau})\|^2$ and $\| \sum_{\bar\tau=t}^{\tau-1} \gamma_{\bar\tau}(\nabla_y g(x_{\bar\tau}, y_{\bar\tau}, \zeta_{\bar\tau,1}) - \nabla_y g(x_t, y_t))\|^2$ are bounded by Eq. (15), Eq. (16) and Eq. (17).*

*Proof.* First for $\| \sum_{\bar\tau=t}^{\tau-1} \eta_{\bar\tau}(\nu_{\bar\tau} - \nabla h(x_t))\|^2$, we have:

$$\Big\| \sum_{\bar\tau=t}^{\tau-1} \eta_{\bar\tau}\big(\nabla h(x_t) - \nu_{\bar\tau}\big) \Big\|^2$$

$$= \Big\| \sum_{\bar\tau=t}^{\tau-1} \eta_{\bar\tau}\big(\nabla_x f(x_t, y_{x_t}) - \nabla_{xy} g(x_t, y_{x_t})u_{x_t} - (\nabla_x f(x_{\bar\tau}, y_{\bar\tau}; \xi_{\bar\tau,2}) - \nabla_{xy} g(x_{\bar\tau}, y_{\bar\tau}; \zeta_{\bar\tau,3})u_{\bar\tau})\big) \Big\|^2$$

$$\leq 2\Big\| \sum_{\bar\tau=t}^{\tau-1} \eta_{\bar\tau}\big(\nabla_x f(x_t, y_{x_t}) - \nabla_x f(x_{\bar\tau}, y_{\bar\tau}; \xi_{\bar\tau,2})\big) \Big\|^2 + 2\Big\| \sum_{\bar\tau=t}^{\tau-1} \eta_{\bar\tau}\big(\nabla_{xy} g(x_t, y_{x_t})u_{x_t} - \nabla_{xy} g(x_{\bar\tau}, y_{\bar\tau}; \zeta_{\bar\tau,3})u_{\bar\tau}\big) \Big\|^2$$

$$\leq 2\Big\| \sum_{\bar\tau=t}^{\tau-1} \eta_{\bar\tau}\big(\nabla_x f(x_t, y_{x_t}) - \nabla_x f(x_{\bar\tau}, y_{\bar\tau}; \xi_{\bar\tau,2})\big) \Big\|^2$$

$$+ 4\Big\| \sum_{\bar\tau=t}^{\tau-1} \eta_{\bar\tau}\nabla_{xy} g(x_{\bar\tau}, y_{\bar\tau}; \zeta_{\bar\tau,3})\big(u_{x_t} - u_{\bar\tau}\big) \Big\|^2 + \frac{4C_f^2}{\mu^2}\Big\| \sum_{\bar\tau=t}^{\tau-1} \eta_{\bar\tau}\big(\nabla_{xy} g(x_t, y_{x_t}) - \nabla_{xy} g(x_{\bar\tau}, y_{\bar\tau}; \zeta_{\bar\tau,3})\big) \Big\|^2$$

$$\leq 2\Big\| \sum_{\bar\tau=t}^{\tau-1} \eta_{\bar\tau}\big(\nabla_x f(x_t, y_{x_t}) - \nabla_x f(x_{\bar\tau}, y_{\bar\tau}; \xi_{\bar\tau,2})\big) \Big\|^2 + 4L^2\bar\eta_t^\tau \sum_{\bar\tau=t}^{\tau-1} \eta_{\bar\tau}\|(u_{x_t} - u_{\bar\tau})\|^2$$

$$+ \frac{4C_f^2}{\mu^2}\Big\| \sum_{\bar\tau=t}^{\tau-1} \eta_{\bar\tau}\big(\nabla_{xy} g(x_t, y_{x_t}) - \nabla_{xy} g(x_{\bar\tau}, y_{\bar\tau}; \zeta_{\bar\tau,3})\big) \Big\|^2 \qquad (12)$$

For the first and the third terms above, we have:

$$\Big\| \sum_{\bar\tau=t}^{\tau-1} \eta_{\bar\tau}\big(\nabla_x f(x_t, y_{x_t}) - \nabla_x f(x_{\bar\tau}, y_{\bar\tau}; \xi_{\bar\tau,2})\big) \Big\|^2$$

$$\leq 3L^2(\bar\eta_t)^2\|y_{x_t} - y_t\|^2 + 3L^2\bar\eta_t \sum_{\bar\tau=t}^{\tau-1} \eta_{\bar\tau}\|z_t - z_{\bar\tau}\|^2 + 3\Big\| \sum_{\bar\tau=t}^{\tau-1} \eta_{\bar\tau}\big(\nabla_x f(x_t, y_t) - \nabla_x f(x_t, y_t; \xi_{\bar\tau,2})\big) \Big\|^2$$

$$\qquad (13)$$

and

$$\|\sum_{\bar\tau=t}^{\tau-1}\eta_{\bar\tau}\big(\nabla_{xy}g(x_t,y_{x_t})-\nabla_{xy}g(x_{\bar\tau},y_{\bar\tau};\zeta_{\bar\tau,3})\big)\|^2$$

$$\leq 3L_{xy}^2(\bar\eta_t^\tau)^2\|y_{x_t}-y_t\|^2+3L_{xy}^2\bar\eta_t^\tau\sum_{\bar\tau=t}^{\tau-1}\eta_{\bar\tau}\|z_t-z_{\bar\tau}\|^2+3\|\sum_{\bar\tau=t}^{\tau-1}\eta_{\bar\tau}\big(\nabla_{xy}g(x_t,y_t)-\nabla_{xy}g(x_t,y_t;\zeta_{\bar\tau,3})\big)\|^2$$

$$(14)$$

Combine Eq (13) and (14) with Eq. (12) and denote $\tilde L_1^2=\big(L^2+\frac{2L_{xy}^2C_f^2}{\mu^2}\big)$ to have:

$$\|\sum_{\bar\tau=t}^{\tau-1}\eta_{\bar\tau}\big(\nu_{\bar\tau}-\nabla h(x_t)\big)\|^2$$

$$\leq 6\tilde L_1^2(\bar\eta_t^\tau)^2\|y_{x_t}-y_t\|^2+6\tilde L_1^2\bar\eta_t^\tau\sum_{\bar\tau=t}^{\tau-1}\eta_{\bar\tau}\|z_t-z_{\bar\tau}\|^2+8L^2(\bar\eta_t^\tau)^2\|u_{x_t}-u_t\|^2+8L^2\bar\eta_t^\tau\sum_{\bar\tau=t}^{\tau-1}\eta_{\bar\tau}\|u_t-u_{\bar\tau}\|^2$$

$$+6\|\sum_{\bar\tau=t}^{\tau-1}\eta_{\bar\tau}\big(\nabla_x f(x_t,y_t)-\nabla_x f(x_t,y_t;\xi_{\bar\tau,2})\big)\|^2+\frac{12C_f^2}{\mu^2}\|\sum_{\bar\tau=t}^{\tau-1}\eta_{\bar\tau}\big(\nabla_{xy}g(x_t,y_t)-\nabla_{xy}g(x_t,y_t;\zeta_{\bar\tau,3})\big)\|^2$$

$$(15)$$

Next, for the term $\|\sum_{\bar\tau=t}^{\tau-1}\rho_{\bar\tau}(\nabla r_{x_t}(u_t)-q_{\bar\tau})\|^2$, we have:

$$\|\sum_{\bar\tau=t}^{\tau-1}\rho_{\bar\tau}\big(\nabla r_{x_t}(u_t)-q_{\bar\tau}\big)\|^2$$

$$\leq\|\sum_{\tau=t}^{\tau-1}\rho_{\bar\tau}\big(\nabla_{y^2}g(x_{\bar\tau},y_{\bar\tau},\zeta_{\bar\tau,2})u_{\bar\tau}-\nabla_y f(x_{\bar\tau},y_{\bar\tau},\xi_{\bar\tau,1})\big)-\big(\nabla_{y^2}g(x_t,y_{x_t})u_t-\nabla_y f(x_t,y_{x_t})\big)\|^2$$

$$\leq 2\|\sum_{\bar\tau=t}^{\tau-1}\rho_{\bar\tau}\big(\nabla_y f(x_t,y_{x_t})-\nabla_y f(x_{\bar\tau},y_{\bar\tau};\xi_{\bar\tau,1})\big)\|^2$$

$$+2\|\sum_{\bar\tau=t}^{\tau-1}\rho_{\bar\tau}\big(\nabla_{y^2}g(x_t,y_{x_t})u_t-\nabla_{y^2}g(x_{\bar\tau},y_{\bar\tau};\zeta_{\bar\tau,2})u_{\bar\tau}\big)\|^2$$

$$\leq 2\|\sum_{\bar\tau=t}^{\tau-1}\rho_{\bar\tau}\big(\nabla_y f(x_t,y_{x_t})-\nabla_y f(x_{\bar\tau},y_{\bar\tau};\xi_{\bar\tau,1})\big)\|^2$$

$$+4\|\sum_{\bar\tau=t}^{\tau-1}\rho_{\bar\tau}\nabla_{y^2}g(x_{\bar\tau},y_{\bar\tau};\zeta_{\bar\tau,2})\big(u_t-u_{\bar\tau}\big)\|^2+\frac{4C_f^2}{\mu^2}\|\sum_{\bar\tau=t}^{\tau-1}\rho_{\bar\tau}\big(\nabla_{y^2}g(x_t,y_{x_t})-\nabla_{y^2}g(x_{\bar\tau},y_{\bar\tau};\zeta_{\tau,2})\big)\|^2$$

$$\leq 2\|\sum_{\bar\tau=t}^{\tau-1}\rho_{\bar\tau}\big(\nabla_y f(x_t,y_{x_t})-\nabla_y f(x_{\bar\tau},y_{\bar\tau};\xi_{\bar\tau,1})\big)\|^2+4L^2\bar\rho_t^\tau\sum_{\bar\tau=t}^{\tau-1}\rho_{\bar\tau}\|(u_t-u_{\bar\tau})\|^2$$

$$+\frac{4C_f^2}{\mu^2}\|\sum_{\bar\tau=t}^{\tau-1}\rho_{\bar\tau}\big(\nabla_{y^2}g(x_t,y_{x_t})-\nabla_{y^2}g(x_{\bar\tau},y_{\bar\tau};\zeta_{\tau,2})\big)\|^2$$

The first and the third terms can be bounded similarly as in Eq. (13) and Eq. (14), and denote $\tilde{L}_2^2 = \left(L^2 + \frac{2L_{y^2}^2 C_f^2}{\mu^2}\right)$, then we have:

$$\|\sum_{\bar\tau=t}^{\tau-1}\rho_{\bar\tau}\big(\nabla r_{x_t}(u_t) - q_{\bar\tau}\big)\|^2$$

$$\leq 6\tilde{L}_2^2(\bar\rho_t^\tau)^2\|y_{x_t} - y_t\|^2 + 6\tilde{L}_2^2\bar\rho_t^\tau \sum_{\bar\tau=t}^{\tau-1}\rho_{\bar\tau}\|z_t - z_{\bar\tau}\|^2 + 4L^2\bar\rho_t^\tau \sum_{\bar\tau=t}^{\tau-1}\rho_{\bar\tau}\|u_t - u_{\bar\tau}\|^2$$

$$+ 6\|\sum_{\bar\tau=t}^{\tau-1}\rho_{\bar\tau}\big(\nabla_y f(x_t, y_t) - \nabla_y f(x_t, y_t; \xi_{\bar\tau,1})\big)\|^2 + \frac{12C_f^2}{\mu^2}\|\sum_{\bar\tau=t}^{\tau-1}\rho_{\bar\tau}\big(\nabla_{y^2} g(x_t, y_t) - \nabla_{y^2} g(x_t, y_t; \zeta_{\bar\tau,2})\big)\|^2$$

$$\tag{16}$$

Finally, for $\|\sum_{\bar\tau=t}^{\tau-1}\gamma_{\bar\tau}(\nabla_y g(x_{\bar\tau}, y_{\bar\tau}, \zeta_{\bar\tau,1}) - \nabla_y g(x_t, y_t))\|^2$, we have:

$$\|\sum_{\bar\tau=t}^{\tau-1}\gamma_{\bar\tau}\big(\nabla_y g(x_{\bar\tau}, y_{\bar\tau}, \zeta_{\bar\tau,1}) - \nabla_y g(x_t, y_t)\big)\|^2$$

$$\leq 2\|\sum_{\bar\tau=t}^{\tau-1}\gamma_{\bar\tau}\big(\nabla_y g(x_{\bar\tau}, y_{\bar\tau}, \zeta_{\bar\tau,1}) - \nabla_y g(x_t, y_t, \zeta_{\bar\tau,1})\big)\|^2$$

$$+ 2\|\sum_{\bar\tau=t}^{\tau-1}\gamma_{\bar\tau}\big(\nabla_y g(x_t, y_t, \zeta_{\bar\tau,1}) - \nabla_y g(x_t, y_t)\big)\|^2$$

$$\leq 2L^2\bar\gamma_t^\tau \sum_{\bar\tau=t}^{\tau-1}\gamma_{\bar\tau}\|z_{\bar\tau} - z_t\|^2 + 2\|\sum_{\bar\tau=t}^{\tau-1}\gamma_{\bar\tau}\big(\nabla_y g(x_t, y_t) - \nabla_y g(x_t, y_t, \zeta_{\bar\tau,1})\big)\|^2 \tag{17}$$

This completes the proof. $\qquad\square$

**Lemma B.9.** *For all $t \geq 0$ and any constant $m > 0$, suppose $\bar\eta_t < \frac{1}{2\bar{L}}$, the iterates generated satisfy:*

$$h(x_{t+k}) \leq h(x_t) - \frac{\bar\eta_t}{2}\|\nabla h(x_t)\|^2 - \frac{\bar\eta_t}{4}\|\bar\nu_t\|^2 + 3\tilde{L}_1^2\bar\eta_t\|y_{x_t} - y_t\|^2 + 3\tilde{L}_1^2 \sum_{\tau=t}^{t+k-1}\eta_\tau\|z_t - z_\tau\|^2$$

$$+ 4L^2\bar\eta_t\|u_{x_t} - u_t\|^2 + 4L^2 \sum_{\tau=t}^{t+k-1}\eta_\tau\|u_t - u_\tau\|^2 + \frac{3}{\bar\eta_t}\|\sum_{\tau=t}^{t+k-1}\eta_\tau\big(\nabla_x f(x_t, y_t) - \nabla_x f(x_t, y_t; \xi_{\tau,2})\big)\|^2$$

$$+ \frac{6C_f^2}{\bar\eta_t\mu^2}\|\sum_{\tau=t}^{t+k-1}\eta_\tau\big(\nabla_{xy} g(x_t, y_t) - \nabla_{xy} g(x_t, y_t; \zeta_{\tau,3})\big)\|^2$$

*Proof.* By the smoothness of $f$, we have:

$$h(x_{t+k}) \leq h(x_t) + \langle\nabla h(x_t), x_{t+k} - x_t\rangle + \frac{\bar{L}}{2}\|x_{t+k} - x_t\|^2$$

$$= h(x_t) - \bar\eta_t\langle\nabla h(x_t), \bar\nu_t\rangle + \frac{\bar\eta_t^2\bar{L}}{2}\|\bar\nu_t\|^2$$

$$\overset{(a)}{=} h(x_t) - \frac{\bar\eta_t}{2}\|\nabla h(x_t)\|^2 + \frac{\bar\eta_t}{2}\|\nabla h(x_t) - \bar\nu_t\|^2 - \left(\frac{\bar\eta_t}{2} - \frac{\bar\eta_t^2\bar{L}}{2}\right)\|\bar\nu_t\|^2$$

$$\overset{(b)}{\leq} h(x_t) - \frac{\bar\eta_t}{2}\|\nabla h(x_t)\|^2 + \frac{\bar\eta_t}{2}\|\nabla h(x_t) - \bar\nu_t\|^2 - \frac{\bar\eta_t}{4}\|\bar\nu_t\|^2 \tag{18}$$

where equality $(a)$ uses $\langle a, b \rangle = \frac{1}{2}[\|a\|^2 + \|b\|^2 - \|a - b\|^2]$; (b) follows the assumption that $\bar{\eta}_t < 1/2\bar{L}$. Next, for the third term, by using Eq. (15), we have:

$$\frac{\bar{\eta}_t}{2}\|\nabla h(x_t) - \bar{\nu}_t\|^2 \leq 3\tilde{L}_1^2\bar{\eta}_t\|y_{x_t} - y_t\|^2 + 3\tilde{L}_1^2 \sum_{\tau=t}^{t+k-1} \eta_\tau\|z_t - z_\tau\|^2$$

$$+ 4L^2\bar{\eta}_t\|u_{x_t} - u_t\|^2 + 4L^2 \sum_{\tau=t}^{t+k-1} \eta_\tau\|u_t - u_\tau\|^2$$

$$+ \frac{3}{\bar{\eta}_t}\| \sum_{\tau=t}^{t+k-1} \eta_\tau\big(\nabla_x f(x_t, y_t) - \nabla_x f(x_t, y_t; \xi_{\tau,2})\big)\|^2$$

$$+ \frac{6C_f^2}{\bar{\eta}_t\mu^2}\| \sum_{\tau=t}^{t+k-1} \eta_\tau\big(\nabla_{xy} g(x_t, y_t) - \nabla_{xy} g(x_t, y_t; \zeta_{\tau,3})\big)\|^2$$

where $\tilde{L}_1^2 = \big(L^2 + \frac{2L_{xy}^2 C_f^2}{\mu^2}\big)$. This completes the proof. $\qquad\square$

**Lemma B.10.** *When $\bar{\gamma}_t < \frac{1}{4L}$, the error of the inner variable $y_t$ are bounded through Eq. (19).*

*Proof.* then by proposition B.19 and choose $\bar{\gamma}_t < \frac{1}{4L}$, we have:

$$\|y_{t+k} - y_{x_t}\|^2 \leq (1 - \frac{\mu\bar{\gamma}_t}{2})\|y_t - y_{x_t}\|^2 - \frac{\bar{\gamma}_t^2}{2}\|\nabla_y g(x_t, y_t)\|^2 + \frac{4\bar{\gamma}_t}{\mu}\|\nabla_y g(x_t, y_t) - \bar{w}_t\|^2$$

Furthermore, by the generalized triangle inequality, we have:

$$\|y_{t+k} - y_{x_{t+k}}\|^2 \leq (1 + \frac{\mu\bar{\gamma}_t}{4})\|y_{t+k} - y_{x_t}\|^2 + (1 + \frac{4}{\mu\bar{\gamma}_t})\|y_{x_{t+k}} - y_{x_t}\|^2$$

$$\leq (1 + \frac{\mu\bar{\gamma}_t}{4})\|y_{t+k} - y_{x_t}\|^2 + \frac{5\kappa^2}{\mu\bar{\gamma}_t}\|x_t - x_{t+k}\|^2$$

$$\leq (1 - \frac{\mu\bar{\gamma}_t}{4})\|y_t - y_{x_t}\|^2 - \frac{\bar{\gamma}_t^2}{2}\|\nabla_y g(x_t, y_t)\|^2 + \frac{5\kappa^2}{\mu\bar{\gamma}_t}\|x_t - x_{t+k}\|^2$$

$$+ \frac{10L^2}{\mu} \sum_{\tau=t}^{t+k-1} \gamma_\tau\|z_t - z_\tau\|^2 + \frac{10}{\mu\bar{\gamma}_t}\| \sum_{\tau=t}^{t+k-1} \gamma_\tau\big(\nabla_y g(x_t, y_t) - \nabla_y g(x_t, y_t; \zeta_{\tau,1})\big)\|^2$$

$$\tag{19}$$

where the second inequality is due to $\bar{\gamma}_t < \frac{1}{4L} < \frac{1}{\mu}$ and uses Eq. (17). This completes the proof. $\quad\square$

**Lemma B.11.** *When $\bar{\rho}_t < \frac{1}{4L}$, the error of variable $u_t$ are bounded through Eq. (20).*

*Proof.* suppose we choose $\bar{\rho}_t < \frac{1}{4L}$, then we have:

$$\|u_{t+k} - u_{x_t}\|^2 \leq (1 - \frac{\mu\bar{\rho}_t}{2})\|u_t - u_{x_t}\|^2 - \frac{(\bar{\rho}_t)^2}{2}\|\nabla r_{x_t}(u_t)\|^2 + \frac{4\bar{\rho}_t}{\mu}\|\nabla r_{x_t}(u_t) - \bar{q}_t\|^2$$

Furthermore, by the generalized triangle inequality, we have:

$$\|u_{t+k} - u_{x_{t+k}}\|^2 \le (1 + \frac{\mu\bar{\rho}_t}{4})\|u_{t+k} - u_{x_t}\|^2 + (1 + \frac{4}{\mu\bar{\rho}_t})\|u_{x_{t+k}} - u_{x_t}\|^2$$

$$\le (1 + \frac{\mu\bar{\rho}_t}{4})\|u_{t+k} - u_{x_t}\|^2 + \frac{5\hat{L}^2}{\mu\bar{\rho}_t}\|x_t - x_{t+k}\|^2$$

$$\le (1 - \frac{\mu\bar{\rho}_t}{4})\|u_t - u_{x_t}\|^2 - \frac{\bar{\rho}_t^2}{2}\|\nabla r_{x_t}(u_t)\|^2 + \frac{5\hat{L}^2}{\mu\bar{\rho}_t}\|x_t - x_{t+k}\|^2$$

$$+ \frac{30\tilde{L}_2^2\bar{\rho}_t}{\mu}\|y_{x_t} - y_t\|^2 + \frac{30\tilde{L}_2^2}{\mu}\sum_{\tau=t}^{t+k-1}\rho_\tau\|z_t - z_\tau\|^2 + \frac{20L^2}{\mu}\sum_{\tau=t}^{t+k-1}\rho_\tau\|u_t - u_\tau\|^2$$

$$+ \frac{30}{\mu\bar{\rho}_t}\|\sum_{\tau=t}^{t+k-1}\rho_\tau\big(\nabla_y f(x_t, y_t) - \nabla_y f(x_t, y_t; \xi_{\tau,1})\big)\|^2$$

$$+ \frac{60C_f^2}{\mu^3\bar{\rho}_t}\|\sum_{\tau=t}^{t+k-1}\rho_\tau\big(\nabla_{y^2} g(x_t, y_t) - \nabla_{y^2} g(x_t, y_t; \zeta_{\tau,2})\big)\|^2 \qquad (20)$$

where in the last inequality, we use the condition that $\bar{\rho}_t < \frac{1}{4L}$ and Eq. (16). This completes the proof. $\qquad\square$

**Lemma B.12.** *Suppose* $\bar{\eta}_t \le \min\left(\frac{1}{96\tilde{L}_1^2}, \frac{1}{32L^2 c_1^2}, \frac{1}{128L^2}, \frac{1}{64L^2 c_2^2}, \frac{1}{96\tilde{L}_2^2 c_2^2}\right)$, *then the drift of* $z_\tau = (x_\tau, y_\tau)$ *and* $u_\tau$ *can be bounded as in Eq. (25) and Eq. (24).*

*Proof.* We first have:

$$\|x_t - x_\tau\|^2 = \|\sum_{\bar{\tau}=t}^{\tau-1}\eta_{\bar{\tau}}\nu_{\bar{\tau}}\|^2 \le 2\|\sum_{\bar{\tau}=t}^{\tau-1}\eta_{\bar{\tau}}\big(\nu_{\bar{\tau}} - \nabla h(x_t)\big)\|^2 + 2(\bar{\eta}_t^\tau)^2\|\nabla h(x_t)\|^2$$

$$\|y_t - y_\tau\|^2 = \|\sum_{\bar{\tau}=t}^{\tau-1}\gamma_{\bar{\tau}}\omega_{\bar{\tau}}\|^2 \le 2\|\sum_{\bar{\tau}=t}^{\tau-1}\gamma_{\bar{\tau}}\big(\nabla_y g(x_{\bar{\tau}}, y_{\bar{\tau}}, \zeta_{\bar{\tau},1}) - \nabla_y g(x_t, y_t)\big)\|^2 + 2(\bar{\gamma}_t^\tau)^2\|\nabla_y g(x_t, y_t)\|^2$$

$$\|u_t - u_\tau\|^2 = \|\sum_{\bar{\tau}=t}^{\tau-1}\rho_{\bar{\tau}}q_{\bar{\tau}}\|^2 \le 2\|\sum_{\bar{\tau}=t}^{\tau-1}\rho_{\bar{\tau}}\big(q_{\bar{\tau}} - \nabla r_{x_t}(u_t)\big)\|^2 + 2(\bar{\rho}_t^\tau)^2\|\nabla r_{x_t}(u_t)\|^2 \qquad (21)$$

For the term $\|z_t - z_\tau\|^2$, we have $\|z_t - z_\tau\|^2 = \|x_t - x_\tau\|^2 + \|y_t - y_\tau\|^2$. Combine Eq. (21) with Eq. (15) and (17), sum $\tau$ over $[t, t+k-1]$ and use Proposition B.18 to have:

$$\sum_{\tau=t}^{t+k-1}\eta_\tau\|z_t - z_\tau\|^2 \le \sum_{\tau=t}^{t+k-1}\eta_\tau\Big(\sum_{\bar{\tau}=t}^{\tau-1}\big(12\tilde{L}_1^2\bar{\eta}_t^\tau\eta_{\bar{\tau}} + 4L^2\bar{\gamma}_t^\tau\gamma_{\bar{\tau}}\big)\|z_t - z_{\bar{\tau}}\|^2\Big)$$

$$+ 2\bar{\alpha}_t\bar{\eta}_t^2\|\nabla h(x_t)\|^2 + 2\bar{\alpha}_t\bar{\gamma}_t^2\|\nabla_y g(x_t, y_t)\|^2$$

$$+ 12\tilde{L}_1^2\bar{\alpha}_t(\bar{\eta}_t)^2\|y_{x_t} - y_t\|^2 + 16L^2\bar{\alpha}_t(\bar{\eta}_t)^2\|u_{x_t} - u_t\|^2$$

$$+ 16L^2\bar{\eta}_t\sum_{\tau=t}^{t+k-1}\eta_\tau\Big(\sum_{\bar{\tau}=t}^{\tau-1}\eta_{\bar{\tau}}\|u_t - u_{\bar{\tau}}\|^2\Big)$$

$$+ 12\bar{\eta}_t\eta_t^2 k^{2-\alpha}C^2 + \frac{24C_f^2}{\mu^2}\bar{\eta}_t\eta_t^2 k^{2-\alpha}C^2 + 4\bar{\eta}_t\gamma_t^2 k^{2-\alpha}C^2$$

By the fact that $\tau \leq t + k$, we have:

$$\sum_{\tau=t}^{t+k-1} \eta_\tau \|z_t - z_\tau\|^2 \leq \sum_{\tau=t}^{t+k-1} \left(12\tilde{L}_1^2(\bar{\eta}_t)^2 \eta_\tau + 4L^2 \bar{\eta}_t \bar{\gamma}_t \gamma_\tau\right) \|z_t - z_\tau\|^2$$
$$+ 2\bar{\eta}_t^3 \|\nabla h(x_t)\|^2 + 2\bar{\eta}_t \bar{\gamma}_t^2 \|\nabla_y g(x_t, y_t)\|^2$$
$$+ 12\tilde{L}_1^2(\bar{\eta}_t)^3 \|y_{x_t} - y_t\|^2 + 16L^2(\bar{\eta}_t)^3 \|u_{x_t} - u_t\|^2$$
$$+ 16L^2(\bar{\eta}_t)^2 \sum_{\tau=t}^{t+k-1} \eta_\tau \|u_t - u_\tau\|^2$$
$$+ 12\bar{\eta}_t \eta_t^2 k^{2-\alpha} C^2 + \frac{24C_f^2}{\mu^2} \bar{\eta}_t \eta_t^2 k^{2-\alpha} C^2 + 4\bar{\eta}_t \gamma_t^2 k^{2-\alpha} C^2 \quad (22)$$

For $\|u_t - u_\tau\|^2$, we combine Eq. (21) with Eq. (16) and sum $\tau$ over $[t, t + k - 1]$ and use the fact that $\tau \leq t + k$ to have:

$$\sum_{\tau=t}^{t+k-1} \eta_\tau \|u_t - u_\tau\|^2 \leq 2\bar{\eta}_t (\bar{\rho}_t)^2 \|\nabla r_{x_t}(u_t)\|^2 + 12\tilde{L}_2^2 \bar{\eta}_t (\bar{\rho}_t)^2 \|y_{x_t} - y_t\|^2$$
$$+ 12\tilde{L}_2^2 \bar{\eta}_t \bar{\rho}_t \sum_{\tau=t}^{t+k-1} \rho_\tau \|z_t - z_\tau\|^2 + 8L^2 \bar{\eta}_t \bar{\rho}_t \sum_{\tau=t}^{t+k-1} \rho_\tau \|u_t - u_\tau\|^2$$
$$+ 12\bar{\eta}_t \rho_t^2 k^{2-\alpha} C^2 + \frac{24C_f^2 \bar{\eta}_t}{\mu^2} \rho_t^2 k^{2-\alpha} C^2 \quad (23)$$

Suppose we have $\gamma_\tau = c_1 \eta_\tau = \frac{20\kappa \tilde{L}_3}{\mu} \eta_\tau$, $\rho_\tau = c_2 \eta_\tau = 40\kappa \hat{L} \eta_\tau$ and set:

$$(\bar{\eta}_t)^2 < \min\left(\frac{1}{96\tilde{L}_1^2}, \frac{1}{32L^2 c_1^2}, \frac{1}{128L^2}, \frac{1}{64L^2 c_2^2}, \frac{1}{96\tilde{L}_2^2 c_2^2}\right)$$

Then we combine Eq. (22) and Eq. (23) to have:

$$\sum_{\tau=t}^{t+k-1} \eta_\tau \|u_t - u_\tau\|^2 \leq 4c_2^2(\bar{\eta}_t)^3 \|\nabla r_{x_t}(u_t)\|^2 + 4\bar{\eta}_t^3 \|\nabla h(x_t)\|^2 + 4c_1^2 \bar{\eta}_t^3 \|\nabla_y g(x_t, y_t)\|^2$$
$$+ \left(24 + \frac{48C_f^2}{\mu^2} + 8c_1^2 + 24c_2^2 + \frac{48c_2^2 C_f^2}{\mu^2}\right) \bar{\eta}_t \eta_t^2 k^{2-\alpha} C^2 \quad (24)$$

and

$$\sum_{\tau=t}^{t+k-1} \eta_\tau \|z_t - z_\tau\|^2 \leq 4c_2^2(\bar{\eta}_t)^3 \|\nabla r_{x_t}(u_t)\|^2 + 4\bar{\eta}_t^3 \|\nabla h(x_t)\|^2 + 4c_1^2 \bar{\eta}_t^3 \|\nabla_y g(x_t, y_t)\|^2$$
$$+ \left(24 + \frac{48C_f^2}{\mu^2} + 8c_1^2 + 24c_2^2 + \frac{48c_2^2 C_f^2}{\mu^2}\right) \bar{\eta}_t \eta_t^2 k^{2-\alpha} C^2 \quad (25)$$

This completes the proof. $\qquad\square$

Suppose we define the potential function $\mathcal{G}(t)$:

$$\mathcal{G}_t = h(x_t) + \phi_y \|y_t - y_{x_t}\|^2 + \phi_u \|u_t - u_{x_t}\|^2$$

where $\phi_u = \frac{32L^2 \bar{\eta}_t}{\mu \bar{\rho}_t}$, $\phi_y = \frac{8\tilde{L}_3^2 \bar{\eta}_t}{\mu \bar{\gamma}_t}$ and $\tilde{L}_3^2 = 960\kappa^2 \tilde{L}_2^2 + 3\tilde{L}_1^2$.

**Theorem B.13.** *Suppose Assumptions 4.1-4.4 are satisfied, and Assumption 4.5 is satisfied with some $\alpha$ and $C$ for the example order we use in Algorithm 1. We choose learning rates $\eta_t = \eta$, $\gamma = c_1 \eta$, $\rho = c_2 \eta$ and denote $T = R \times I$ be the total number of steps. Then for any pair of values $(E, k)$ which has $T = E \times k$ and $\eta \leq \frac{1}{k\tilde{L}_0}$, we have:*

$$\frac{1}{E} \sum_{e=0}^{E-1} \|\nabla h(x_{ke})\|^2 \leq \frac{2\Delta}{kE\eta} + 2\tilde{L}^2 k^{-\alpha} C^2$$

*where $c_1$, $c_2$, $\tilde{L}_0$, $\tilde{L}$ are constants related to the smoothness parameters of $h(x)$, $\Delta$ is the initial sub-optimality, $R$ and $I$ are defined in Algorithm 1.*

*Proof.* First we combine Lemma B.9-Lemma B.11, and use the condition that $\gamma_t = \frac{20\kappa\tilde{L}_3}{\mu}\eta_t$, $\rho_t = 40\kappa\hat{L}\eta_t$, and by the assumption about gradient error:

$$
\begin{aligned}
\mathcal{G}_{t+k} - \mathcal{G}_t \leq{} & -\frac{\bar{\eta}_t}{2}\|\nabla h(x_t)\|^2 - \frac{4c_1\tilde{L}_3^2(\bar{\eta}_t)^2}{\mu}\|\nabla_y g(x_t, y_t)\|^2 - \frac{16c_2 L^2(\bar{\eta}_t)^2}{\mu}\|\nabla r_{x_t}(u_t)\|^2 \\
& - \tilde{L}_3^2\bar{\eta}_t\|y_{x_t} - y_t\|^2 - 4L^2\bar{\eta}_t\|u_{x_t} - u_t\|^2 \\
& + \sum_{\tau=t}^{t+k-1}\left(960\kappa^2\tilde{L}_2^2\eta_\tau + 3\tilde{L}_1^2\eta_\tau + 80\kappa^2\tilde{L}_3^2\eta_\tau\right)\|z_t - z_\tau\|^2 \\
& + \sum_{\tau=t}^{t+k-1}\left(640\kappa^2 L^2\eta_\tau + 4L^2\eta_\tau\right)\|u_t - u_\tau\|^2 \\
& + \left(3 + \frac{6C_f^2}{\mu^2} + \frac{80\tilde{L}_3^2}{\mu^2} + 960\kappa^2 + \frac{1920\kappa^2 C_f^2}{\mu^2}\right)\frac{\eta_t^2 k^{2-\alpha}C^2}{\bar{\eta}_t}
\end{aligned}
\tag{26}
$$

Suppose we denote $\tilde{L}_4^2 = \left(960\kappa^2\tilde{L}_2^2 + 3\tilde{L}_1^2 + 80\kappa^2\tilde{L}_3^2 + 640\kappa^2 L^2 + 4L^2\right)$, and set:

$$
(\bar{\eta}_t)^2 < \min\left(\frac{1}{16\tilde{L}_4^2}, \frac{\tilde{L}_3^4}{c_1^2\mu^2\tilde{L}_4^4}, \frac{16L^4}{c_2^2\mu^2\tilde{L}_4^4}, \frac{\tilde{L}_3^2}{48\tilde{L}_4^2\tilde{L}_2^2 c_2^2}, \frac{\tilde{L}_3^2}{48\tilde{L}_4^2\tilde{L}_1^2}, \frac{1}{8\tilde{L}_4^2}\right)
$$

and we bound the terms $\|u_{x_t} - u_t\|^2$ and $\|z_t - z_\tau\|^2$ through Lemma B.12 to have:

$$
\begin{aligned}
\mathcal{G}_{t+k} - \mathcal{G}_t \leq{} & -\frac{\bar{\eta}_t}{2}\|\nabla h(x_t)\|^2 + \left(24 + \frac{48C_f^2}{\mu^2} + 8c_1^2 + 24c_2^2 + \frac{48c_2^2 C_f^2}{\mu^2}\right)\tilde{L}_4^2\bar{\eta}_t\eta_t^2 k^{2-\alpha}C^2 \\
& + \left(3 + \frac{6C_f^2}{\mu^2} + \frac{80\tilde{L}_3^2}{\mu^2} + 960\kappa^2 + \frac{1920\kappa^2 C_f^2}{\mu^2}\right)\frac{\eta_t^2 k^{2-\alpha}C^2}{\bar{\eta}_t}
\end{aligned}
$$

Next suppose we set $\tilde{L}_4\bar{\eta}_t < 1$, then we have:

$$
\begin{aligned}
\mathcal{G}_{t+k} - \mathcal{G}_t \leq{} & -\frac{\bar{\eta}_t}{2}\|\nabla h(x_t)\|^2 + \Big(24 + \frac{48C_f^2}{\mu^2} + 8c_1^2 + 24c_2^2 + \frac{48c_2^2 C_f^2}{\mu^2} + \\
& 3 + \frac{6C_f^2}{\mu^2} + \frac{80\tilde{L}_3^2}{\mu^2} + 960\kappa^2 + \frac{1920\kappa^2 C_f^2}{\mu^2}\Big)\frac{\eta_t^2 k^{2-\alpha}C^2}{\bar{\eta}_t}
\end{aligned}
$$

For ease of notation we denote $\tilde{L}^2 = \left(24 + \frac{48C_f^2}{\mu^2} + 8c_1^2 + 24c_2^2 + \frac{48c_2^2 C_f^2}{\mu^2} + 3 + \frac{6C_f^2}{\mu^2} + \frac{80\tilde{L}_3^2}{\mu^2} + 960\kappa^2 + \frac{1920\kappa^2 C_f^2}{\mu^2}\right)$, then we have:

$$
\mathcal{G}_{t+k} - \mathcal{G}_t \leq -\frac{\bar{\eta}_t}{2}\|\nabla h(x_t)\|^2 + \frac{\tilde{L}^2\eta_t^2 k^{2-\alpha}C^2}{\bar{\eta}_t}
$$

Sum the above inequality over $E$ phases to have:

$$
\begin{aligned}
\sum_{e=0}^{E-1}\frac{\bar{\eta}_{ke}}{2}\|\nabla h(x_{ke})\|^2 \leq{} & \mathcal{G}_0 - \mathcal{G}_{kE} + \sum_{e=0}^{E-1}\frac{\tilde{L}^2\eta_{ke}^2 k^{2-\alpha}C^2}{\bar{\eta}_{ke}} \\
\leq{} & \Delta_h + \phi_y\Delta_y + \phi_u\Delta_u + \sum_{e=0}^{E-1}\frac{\tilde{L}^2\eta_{ke}^2 k^{2-\alpha}C^2}{\bar{\eta}_{ke}}
\end{aligned}
$$

we define $\Delta_h = h(x_0) - h^*$ as the initial sub-optimality of the function, $\Delta_y = \|y_0 - y_{x_0}\|^2$ as the initial sub-optimality of the inner variable estimation, $\Delta_u = \|u_0 - u_{x_0}\|^2$ as the initial sub-optimality of the hyper-gradient estimation. Then suppose we choose $\eta_t = \eta$ be some constant, then we have:

$$
\frac{k\eta}{2}\sum_{e=0}^{E-1}\|\nabla h(x_{ke})\|^2 \leq \Delta_h + \phi_y\Delta_y + \phi_u\Delta_u + \tilde{L}^2 Ek^{1-\alpha}C^2\eta
$$

Divide by $kE\eta/2$ on both sides to have:

$$\frac{1}{E}\sum_{e=0}^{E-1}\|\nabla h(x_{ke})\|^2 \le \frac{2}{kE\eta}\big(\Delta_h + \phi_y\Delta_y + \phi_u\Delta_u\big) + 2\tilde{L}^2 k^{-\alpha}C^2$$

Suppose we choose the largest $\eta = \frac{1}{k\tilde{L}_0}$ such that the conditions of Lemma B.9-Lemma B.12 are satisfied, then we have:

$$\frac{1}{E}\sum_{e=0}^{E-1}\|\nabla h(x_{ke})\|^2 \le \frac{2\tilde{L}_0}{E}\big(\Delta + \phi_y\Delta_y + \phi_u\Delta_u\big) + 2\tilde{L}^2 k^{-\alpha}C^2$$

Then to reach an $\epsilon$-stationary point, we need to have:

$$E \ge \frac{4\tilde{L}_0\Delta}{\epsilon^2}, \text{ and }, k \ge \left(\frac{4\tilde{L}^2 C^2}{\epsilon^2}\right)^{1/\alpha}$$

in other words $T = Ek \ge \frac{4\tilde{L}_0(4\tilde{L}^2C^2)^{1/\alpha}\Delta}{\epsilon^{2+2/\alpha}}$. This completes the proof. $\qquad\square$

## B.2 Proof for conditional bilevel optimization problems

In this section we study the convergence rate of Algorithm 2.

**Lemma B.14.** *For all $t \ge 0$ and any constant $k > 0$, suppose $\bar{\eta}_t < \frac{1}{4\tilde{L}}$, the iterates generated satisfy:*

$$h(x_{t+k}) \le h(x_t) - \frac{k\eta}{8}\|\nabla h(x_t)\|^2 - \frac{k\eta}{4}\|\bar{\nu}_t\|^2 + 3\big(1 + \frac{4\kappa^2(192C_f^2 + 96*24\tilde{L}_2^2)}{\mu^2} + \frac{48\tilde{L}_1^2}{\mu^2}\big)\eta k^{1-\alpha}C^2$$

$$+ 12\eta k L^2(1 - \frac{k\mu\rho}{2})^{E_l}\Delta_u + 3\eta k(2\tilde{L}_1^2 + 4*96\kappa^2\tilde{L}_2^2)(1 - \frac{k\mu\gamma}{2})^{E_l}\Delta_y$$

*Proof.* By the smoothness of $f$, follow similar derivation as in Eq. (18), we have:

$$h(x_{t+k}) \le h(x_t) - \frac{\bar{\eta}_t}{2}\|\nabla h(x_t)\|^2 + \frac{\bar{\eta}_t}{2}\|\nabla h(x_t) - \bar{\nu}_t\|^2 - \frac{\bar{\eta}_t}{4}\|\bar{\nu}_t\|^2$$

In particular, for the term $\|\sum_{\bar{\tau}=t}^{\tau-1}\eta_{\bar{\tau}}(\nu_{\bar{\tau}} - \nabla h(x_t))\|^2$, we have:

$$\Big\|\sum_{\bar{\tau}=t}^{\tau-1}\eta_{\bar{\tau}}\big(\nabla h(x_t) - \nu_{\bar{\tau}}\big)\Big\|^2$$

$$\le 3\Big\|\sum_{\bar{\tau}=t}^{\tau-1}\eta_{\bar{\tau}}\big(\nabla h(x_t) - \nabla h(x_t;\xi_{\bar{\tau}})\big)\Big\|^2 + 3\Big\|\sum_{\bar{\tau}=t}^{\tau-1}\eta_{\bar{\tau}}\big(\nabla h(x_t;\xi_{\bar{\tau}}) - \nabla h(x_{\bar{\tau}};\xi_{\bar{\tau}})\big)\Big\|^2 + 3\Big\|\sum_{\bar{\tau}=t}^{\tau-1}\eta_{\bar{\tau}}\big(\nabla h(x_{\bar{\tau}};\xi_{\bar{\tau}}) - \nu_{\bar{\tau}}\big)\Big\|^2$$

$$\le 3\Big\|\sum_{\bar{\tau}=t}^{\tau-1}\eta_{\bar{\tau}}\big(\nabla h(x_t) - \nabla h(x_t;\xi_{\bar{\tau}})\big)\Big\|^2 + 3\bar{L}^2\bar{\eta}_t^\tau\sum_{\bar{\tau}=t}^{\tau-1}\eta_{\bar{\tau}}\big\|x_t - x_{\bar{\tau}}\big\|^2 + 3\bar{\eta}_t^\tau\sum_{\bar{\tau}=t}^{\tau-1}\eta_{\bar{\tau}}\big\|\nabla h(x_{\bar{\tau}};\xi_{\bar{\tau}}) - \nu_{\bar{\tau}}\big\|^2$$
$$\tag{27}$$

For the third term, we follow similar derivation as in Eq. (12)-Eq. (15), we have:

$$\big\|\big(\nabla h(x_\tau;\xi_\tau) - \nu_\tau\big)\big\|^2$$
$$= \big\|\big(\nabla_x f(x_\tau, y_{x_\tau}^{\xi_\tau};\xi_\tau) - \nabla_{xy}g^{\xi_\tau}(x_\tau, y_{x_\tau}^{\xi_\tau})u_{x_\tau}^{\xi_\tau} - (\nabla_x f(x_\tau, y_{\tau,T_l};\xi_\tau) - \nabla_{xy}g^{\xi_\tau}(x_\tau, y_{\tau,T_l})u_{\tau,T_l})\big)\big\|^2$$
$$\le 2\tilde{L}_1^2\|y_{x_\tau}^{\xi_\tau} - y_{\tau,T_l}\|^2 + 4L^2\|u_{x_\tau}^{\xi_\tau} - u_{\tau,T_l}\|^2 \tag{28}$$

where $\tilde{L}_1^2 = \big(L^2 + \frac{2L_{xy}^2C_f^2}{\mu^2}\big)$. Combine Eq. (27) and Eq. (28), we have:

$$h(x_{t+k}) \le h(x_t) - \frac{\bar{\eta}_t}{2}\|\nabla h(x_t)\|^2 - \frac{\bar{\eta}_t}{4}\|\bar{\nu}_t\|^2 + \frac{3}{2\bar{\eta}_t}\Big\|\sum_{\tau=t}^{t+k-1}\eta_\tau\big(\nabla h(x_t) - \nabla h(x_t;\xi_\tau)\big)\Big\|^2$$

$$+ \frac{3\bar{L}^2}{2}\sum_{\tau=t}^{t+k-1}\eta_\tau\big\|x_t - x_\tau\big\|^2 + \sum_{\tau=t}^{t+k-1}\eta_\tau\big(3\tilde{L}_1^2\|y_{x_\tau}^{\xi_\tau} - y_{\tau,T_l}\|^2 + 6L^2\|u_{x_\tau}^{\xi_\tau} - u_{\tau,T_l}\|^2\big)$$
$$\tag{29}$$

For the term $\sum_{\tau=t}^{t+k-1}\eta_\tau\|x_t - x_\tau\|^2$, by Eq. (27) and Eq. (28), we have:

$$\|x_t - x_\tau\|^2 \leq 2\|\sum_{\bar{\tau}=t}^{\tau-1}\eta_{\bar{\tau}}\big(\nu_{\bar{\tau}} - \nabla h(x_t)\big)\|^2 + 2(\bar{\eta}_t^\tau)^2\|\nabla h(x_t)\|^2$$

$$\leq 6\|\sum_{\bar{\tau}=t}^{\tau-1}\eta_{\bar{\tau}}\big(\nabla h(x_t) - \nabla h(x_t;\xi_{\bar{\tau}})\big)\|^2 + 6\bar{L}^2\bar{\eta}_t^\tau\sum_{\bar{\tau}=t}^{\tau-1}\eta_{\bar{\tau}}\|x_t - x_{\bar{\tau}}\|^2$$

$$+ 6\bar{\eta}_t^\tau\sum_{\bar{\tau}=t}^{\tau-1}\eta_{\bar{\tau}}\big(2\tilde{L}_1^2\|y_{x_{\bar{\tau}}}^{\xi_{\bar{\tau}}} - y_{\bar{\tau},T_l}\|^2 + 4L^2\|u_{x_{\bar{\tau}}}^{\xi_{\bar{\tau}}} - u_{\bar{\tau},T_l}\|^2\big) + 2(\bar{\eta}_t^\tau)^2\|\nabla h(x_t)\|^2$$

$$\leq 6\eta_t^2 k^{2-\alpha}C^2 + 6\bar{L}^2\bar{\eta}_t\sum_{\bar{\tau}=t}^{t+k-1}\eta_{\bar{\tau}}\|x_t - x_{\bar{\tau}}\|^2$$

$$+ 6\bar{\eta}_t\sum_{\bar{\tau}=t}^{t+k-1}\eta_{\bar{\tau}}\big(2\tilde{L}_1^2\|y_{x_{\bar{\tau}}}^{\xi_{\bar{\tau}}} - y_{\bar{\tau},T_l}\|^2 + 4L^2\|u_{x_{\bar{\tau}}}^{\xi_{\bar{\tau}}} - u_{\bar{\tau},T_l}\|^2\big) + 2(\bar{\eta}_t)^2\|\nabla h(x_t)\|^2$$

Multiply $\eta_\tau$ on both sides and sum over $[t, t+k-1]$, we have:

$$(1 - 6\bar{L}^2(\bar{\eta}_t)^2)\sum_{\tau=t}^{t+k-1}\eta_\tau\|x_t - x_\tau\|^2 \leq 6(\bar{\eta}_t)^2\sum_{\bar{\tau}=t}^{t+k-1}\eta_{\bar{\tau}}\big(2\tilde{L}_1^2\|y_{x_{\bar{\tau}}}^{\xi_{\bar{\tau}}} - y_{\bar{\tau},T_l}\|^2 + 4L^2\|u_{x_{\bar{\tau}}}^{\xi_{\bar{\tau}}} - u_{\bar{\tau},T_l}\|^2\big)$$

$$+ 2(\bar{\eta}_t)^3\|\nabla h(x_t)\|^2 + 6\bar{\eta}_t\eta_t^2 k^{2-\alpha}C^2$$

By the condition that $\bar{\eta}_t < \frac{1}{4L}$, we have:

$$\sum_{\tau=t}^{t+k-1}\eta_\tau\|x_t - x_\tau\|^2 \leq 12(\bar{\eta}_t)^2\sum_{\bar{\tau}=t}^{t+k-1}\eta_{\bar{\tau}}\big(2\tilde{L}_1^2\|y_{x_{\bar{\tau}}}^{\xi_{\bar{\tau}}} - y_{\bar{\tau},T_l}\|^2 + 4L^2\|u_{x_{\bar{\tau}}}^{\xi_{\bar{\tau}}} - u_{\bar{\tau},T_l}\|^2\big)$$

$$+ 4(\bar{\eta}_t)^3\|\nabla h(x_t)\|^2 + 12\bar{\eta}_t\eta_t^2 k^{2-\alpha}C^2 \tag{30}$$

Combine Eq. (30) with Eq. (29), then we have:

$$h(x_{t+k}) \leq h(x_t) - (\frac{\bar{\eta}_t}{2} - 6\bar{L}^2(\bar{\eta}_t)^3)\|\nabla h(x_t)\|^2 - \frac{\bar{\eta}_t}{4}\|\bar{\nu}_t\|^2 + \frac{3}{2\bar{\eta}_t}\|\sum_{\tau=t}^{t+k-1}\eta_\tau\big(\nabla h(x_t) - \nabla h(x_t;\xi_\tau)\big)\|^2$$

$$+ 18\bar{L}^2\bar{\eta}_t\eta_t^2 k^{2-\alpha}C^2 + (1 + 12(\bar{\eta}_t)^2\bar{L}^2)\sum_{\tau=t}^{t+k-1}\eta_\tau\big(3\tilde{L}_1^2\|y_{x_\tau}^{\xi_\tau} - y_{\tau,T_l}\|^2 + 6L^2\|u_{x_\tau}^{\xi_\tau} - u_{\tau,T_l}\|^2\big)$$

$$\leq h(x_t) - \frac{\bar{\eta}_t}{8}\|\nabla h(x_t)\|^2 - \frac{\bar{\eta}_t}{4}\|\bar{\nu}_t\|^2 + \frac{3}{\bar{\eta}_t}\eta_t^2 k^{2-\alpha}C^2$$

$$+ \sum_{\tau=t}^{t+k-1}\eta_\tau\big(6\tilde{L}_1^2\|y_{x_\tau}^{\xi_\tau} - y_{\tau,T_l}\|^2 + 12L^2\|u_{x_\tau}^{\xi_\tau} - u_{\tau,T_l}\|^2\big) \tag{31}$$

where the second inequality uses the conditions that $\bar{\eta}_t < \frac{1}{4L}$. By Lemma B.16, we have:

$$h(x_{t+k}) \leq h(x_t) - \frac{k\eta}{8}\|\nabla h(x_t)\|^2 - \frac{k\eta}{4}\|\bar{\nu}_t\|^2 + 3\big(1 + \frac{4\kappa^2(192C_f^2 + 96*24\tilde{L}_2^2)}{\mu^2} + \frac{48\tilde{L}_1^2}{\mu^2}\big)\eta k^{1-\alpha}C^2$$

$$+ 12\eta k L^2(1 - \frac{k\mu\rho}{2})^{E_l}\Delta_u + 3\eta k(2\tilde{L}_1^2 + 4*96\kappa^2\tilde{L}_2^2)(1 - \frac{k\mu\gamma}{2})^{E_l}\Delta_y \tag{32}$$

where $\Delta_u$ and $\Delta_y$ denotes the upper bounds of the initial estimation errors of $y_x$ and $u_x$. This completes the proof. $\qquad\square$

**Lemma B.15.** *Suppose we have* $\bar{\gamma}_l < \frac{1}{2L}$ *and* $\bar{\rho}_l < \frac{1}{2L}$, *then* $\sum_{l'=l}^{l+k-1}\gamma_{l'}\|y_{l'} - y_l\|^2$ *and* $\sum_{l'=l}^{l+k-1}\rho_{l'}\|u_l - u_{l'}\|^2$ *can be bounded as in Eq. (33) and Eq. (34)*

*Proof.* We first bound $\sum_{l'=l}^{l+k-1} \gamma_{l'} \|y_{l'} - y_l\|^2$, in fact, we have:

$$\|y_l - y_{l'}\|^2 = \|\sum_{\bar{l}'=l}^{l'-1} \gamma_{\bar{l}'} w_{\bar{l}'}\|^2 \le 2\|\sum_{\bar{l}'=l}^{l'-1} \gamma_{\bar{l}'} \left(\nabla_y g^{\zeta_{\bar{l}'}}(x, y_{\bar{l}'}) - \nabla_y g(x, y_l)\right)\|^2 + 2(\bar{\gamma}_l^{l'})^2 \|\nabla_y g(x, y_l)\|^2$$

$$\le 4L^2 \bar{\gamma}_l^{l'} \sum_{\bar{l}'=l}^{l'-1} \gamma_{\bar{l}'} \|y_{\bar{l}'} - y_l\|^2 + 2(\bar{\gamma}_l^{l'})^2 \|\nabla_y g(x, y_l)\|^2 + 4\|\sum_{\bar{l}'=l}^{l'-1} \gamma_{\bar{l}'} \left(\nabla_y g^{\zeta_{\bar{l}'}}(x, y_l) - \nabla_y g(x, y_l)\right)\|^2$$

$$\le 4L^2 \bar{\gamma}_l \sum_{\bar{l}'=l}^{l+k-1} \gamma_{\bar{l}'} \|y_{\bar{l}'} - y_l\|^2 + (\bar{\gamma}_l)^2 \|\nabla_y g(x, y_l)\|^2 + 4\gamma_l^2 k^{2-\alpha} C^2$$

Multiple $\gamma_{l'}$ on both sides, and then sum $l'$ in $[l, l+k-1]$, we have:

$$(1 - 4L^2 \bar{\gamma}_l^2) \sum_{l'=l}^{l+k-1} \gamma_{l'} \|y_{l'} - y_l\|^2 \le 2(\bar{\gamma}_l)^3 \|\nabla_y g(x, y_l)\|^2 + 4\bar{\gamma}_l \gamma_l^2 k^{2-\alpha} C^2$$

Since we have $\bar{\gamma}_l < \frac{1}{2L}$, we have:

$$\sum_{l'=l}^{l+k-1} \gamma_{l'} \|y_{l'} - y_l\|^2 \le 4(\bar{\gamma}_l)^3 \|\nabla_y g(x, y_l)\|^2 + 8\bar{\gamma}_l \gamma_l^2 k^{2-\alpha} C^2 \tag{33}$$

Next for $\sum_{l'=l}^{l+k-1} \rho_{l'} \|u_l - u_{l'}\|^2$, we have:

$$\|u_l - u_{l'}\|^2 = \|\sum_{\bar{l}'=t}^{l'-1} \rho_{\bar{l}'} q_{\bar{l}'}\|^2 \le 2\|\sum_{\bar{l}'=l}^{l'-1} \rho_{\bar{l}'} \left(q_{\bar{l}'} - \nabla r_x(u_l)\right)\|^2 + 2(\bar{\rho}_l^{l'})^2 \|\nabla r_x(u_l)\|^2$$

$$\le 12\tilde{L}_2^2 (\bar{\rho}_l^{l'})^2 \|y_x - y_{T_l}\|^2 + 8L^2 \bar{\rho}_l^{l'} \sum_{\bar{l}'=l}^{l'-1} \rho_{\bar{l}'} \|u_l - u_{\bar{l}'}\|^2 + 2(\bar{\rho}_l^{l'})^2 \|\nabla r_x(u_l)\|^2$$

$$+ \frac{24C_f^2}{\mu^2} \|\sum_{\bar{l}'=l}^{l'-1} \rho_{\bar{l}'} \left(\nabla_{y^2} g(x, y_{T_l}) - \nabla_{y^2} g^{\zeta_{\bar{l}'}}(x, y_{T_l})\right)\|^2$$

$$\le 12\tilde{L}_2^2 (\bar{\rho}_l)^2 \|y_x - y_{T_l}\|^2 + 8L^2 \bar{\rho}_l \sum_{\bar{l}'=l}^{l+k-1} \rho_{\bar{l}'} \|u_l - u_{\bar{l}'}\|^2 + 2(\bar{\rho}_l)^2 \|\nabla r_x(u_l)\|^2 + \frac{24C_f^2}{\mu^2} \rho_l^2 k^{2-\alpha} C^2$$

Multiple $\rho_{l'}$ on both sides, and then sum $l'$ in $[l, l+k-1]$, we have:

$$(1 - 8L^2 \bar{\rho}_l^2) \sum_{l'=l}^{l+k-1} \rho_{l'} \|u_l - u_{l'}\|^2 \le 12\tilde{L}_2^2 (\bar{\rho}_l)^3 \|y_x - y_{T_l}\|^2 + 2(\bar{\rho}_l)^3 \|\nabla r_x(u_l)\|^2 + \frac{24C_f^2}{\mu^2} \bar{\rho}_l \rho_l^2 k^{2-\alpha} C^2$$

Since we have $\bar{\rho}_l < \frac{1}{2L}$, we have:

$$\sum_{l'=l}^{l+k-1} \rho_{l'} \|u_l - u_{l'}\|^2 \le 24\tilde{L}_2^2 (\bar{\rho}_l)^3 \|y_x - y_{T_l}\|^2 + 4(\bar{\rho}_l)^3 \|\nabla r_x(u_l)\|^2 + \frac{48C_f^2}{\mu^2} \bar{\rho}_l \rho_l^2 k^{2-\alpha} C^2 \tag{34}$$

$\square$

**Lemma B.16.** *When $\bar{\gamma}_t < \frac{1}{128L\kappa}$ and $\bar{\rho}_t < \frac{1}{256L\kappa}$, the error of the inner variable $y_l$ and the variable $u_l$ are bounded through Eq. (35) and Eq. (36).*

*Proof.* Since the outer iteration $t$ is fixed for each inner loop, we omit $t$ and $\xi_t$ in the notation for clarity. Furthermore, we assume performing the updates to $y$ and $u$ separately in Algorithm 2. Although the alternative updates in Algorithm 2 is appealing in practice as we can reuse $\nabla_y g$ for

$\nabla_{y^2} g$, update $y$ first and $u$ next can avoid treating complicated higher order terms. Follow similar derivation of Lemma B.10, if $\bar{\gamma}_t < \frac{1}{4L}$, we have:

$$\|y_{l+k} - y_x\|^2 \leq (1 - \frac{\mu\bar{\gamma}_l}{2})\|y_l - y_x\|^2 - \frac{\bar{\gamma}_l^2}{2}\|\nabla_y g(x, y_l)\|^2$$
$$+ 8L\kappa \sum_{l'=l}^{l+k-1} \gamma_{l'}\|y_{l'} - y_l\|^2 + \frac{8}{\mu\bar{\gamma}_l}\|\sum_{l'=l}^{l+k-1} \gamma_{l'}\left(\nabla_y g(x, y_l) - \nabla_y g(x, y_l; \zeta_{l'})\right)\|^2$$

Combine with Eq. (33) and use the condition that $\bar{\gamma}_l < \frac{1}{128L\kappa} < \frac{1}{4L}$, we have:

$$\|y_{l+k} - y_x\|^2 \leq (1 - \frac{\mu\bar{\gamma}_l}{2})\|y_l - y_x\|^2 - \frac{\bar{\gamma}_l^2}{4}\|\nabla_y g(x, y_l)\|^2 + 64L\kappa\bar{\gamma}_l\gamma_l^2 k^{2-\alpha}C^2 + \frac{8}{\mu\bar{\gamma}_l}\gamma_l^2 k^{2-\alpha}C^2$$

Suppose we perform $E_l$ rounds, then by telescoping, we have:

$$\|y_{T_l} - y_x\|^2 \leq (1 - \frac{k\mu\gamma}{2})^{E_l}\|y_0 - y_x\|^2 + \frac{24k^{-\alpha}C^2}{\mu^2} + \sum_{e=0}^{E_l-1}(1 - \frac{k\mu\gamma}{2})^{E_l-e}\left(-\frac{k^2\gamma^2}{4}\|\nabla_y g(x, y_{ek})\|^2\right)$$

$$(35)$$

Next, follow similar derivations as in Lemma B.11 and choose $\bar{\rho}_t < \frac{1}{4L}$, then we have:

$$\|u_{l+k} - u_x\|^2 \leq (1 - \frac{\mu\bar{\rho}_l}{2})\|u_l - u_x\|^2 - \frac{(\bar{\rho}_l)^2}{2}\|\nabla r_x(u_l)\|^2 + \frac{24\tilde{L}_2^2(\bar{\rho}_t)}{\mu}\|y_x - y_{T_l}\|^2$$
$$+ 16L\kappa \sum_{l'=l}^{l+k-1} \rho_{l'}\|u_l - u_{l'}\|^2 + \frac{48C_f^2}{\mu^3\bar{\rho}_l}\|\sum_{l'=l}^{l+k-1} \rho_{l'}\left(\nabla_{y^2} g(x, y_{T_l}) - \nabla_{y^2} g(x, y_{T_l}; \zeta_{l'})\right)\|^2$$

Combine with Eq. (34) and use the condition that $\bar{\rho}_l < \frac{1}{256L\kappa} < \frac{1}{4L}$, we have:

$$\|u_{l+k} - u_x\|^2 \leq (1 - \frac{\mu\bar{\rho}_l}{2})\|u_l - u_x\|^2 - \frac{(\bar{\rho}_l)^2}{4}\|\nabla r_x(u_l)\|^2 + \frac{48\tilde{L}_2^2(\bar{\rho}_l)}{\mu}\|y_x - y_{T_l}\|^2 + \frac{96C_f^2}{\mu^3\bar{\rho}_l}\rho_l^2 k^{2-\alpha}C^2$$

we perform $E_l$ rounds, then by telescoping, we have:

$$\|u_{T_l} - u_x\|^2 \leq (1 - \frac{k\mu\rho}{2})^{E_l}\|u_0 - u_x\|^2 + \frac{192C_f^2C^2k^{-\alpha}}{\mu^4} + \frac{96\tilde{L}_2^2}{\mu^2}\|y_x - y_{T_l}\|^2$$

$$+ \sum_{e=0}^{E_l-1}(1 - \frac{k\mu\rho}{2})^{E_l-e}\left(-\frac{k^2\rho^2}{4}\|\nabla_y r_x(u_{ek})\|^2\right)$$

$$(36)$$

This completes the proof. $\qquad\square$

**Theorem B.17.** *Suppose Assumptions 4.1-4.4 and 4.7 are satisfied, and Assumption 4.5 is satisfied with some $\alpha$ and $C$ for the example order we use in Algorithm 2. We choose learning rates $\eta_t = \eta = \frac{1}{8kL}$, $\gamma_t = \gamma = \frac{1}{256kL\kappa}$, $\rho_t = \rho = \frac{1}{512kL\kappa}$ and denote $T = R \times m$ be the total number of outer steps and $T_l = S \times \max(n_i)$ be the maximum inner steps. Then for any pair of values (E, k) which has $T = E \times k$ and $T_l = E_l \times k$, we have:*

$$\frac{1}{E}\sum_{e=0}^{E-1}\|\nabla h(x_{ke})\|^2 \leq \frac{8\Delta_h}{kE\eta} + C_u(1 - \frac{k\mu\rho}{2})^{E_l}\Delta_u + C_y(1 - \frac{k\mu\gamma}{2})^{E_l}\Delta_y + (C')^2 C^2 k^{-\alpha}$$

*where $C_u$, $C_y$, $C'$, $\tilde{L}$ are constants related to the smoothness parameters of $h(x)$, $\Delta_h$, $\Delta_u$, $\Delta_y$ are the initial sub-optimality, R and S are defined in Algorithm 2.*

*Proof.* First, by Lemma B.14, we have:

$$h(x_{t+k}) \leq h(x_t) - \frac{k\eta}{8}\|\nabla h(x_t)\|^2 - \frac{k\eta}{4}\|\bar{\nu}_t\|^2 + 3\left(1 + \frac{4\kappa^2(192C_f^2 + 96*24\tilde{L}_2^2)}{\mu^2} + \frac{48\tilde{L}_1^2}{\mu^2}\right)\eta k^{1-\alpha}C^2$$

$$+ 12\eta kL^2(1 - \frac{k\mu\rho}{2})^{E_l}\Delta_u + 3\eta k(2\tilde{L}_1^2 + 4*96\kappa^2\tilde{L}_2^2)(1 - \frac{k\mu\gamma}{2})^{E_l}\Delta_y$$

Sum the above inequality over $E$ phases to have:

$$\frac{k\eta}{8}\sum_{e=0}^{E-1}\|\nabla h(x_{ke})\|^2 \leq \Delta_h + 12\eta kEL^2(1-\frac{k\mu\rho}{2})^{E_l}\Delta_u + 3\eta kE(2\tilde{L}_1^2 + 4*96\kappa^2\tilde{L}_2^2)(1-\frac{k\mu\gamma}{2})^{E_l}\Delta_y$$

$$+ 3E\big(1 + \frac{4\kappa^2(192C_f^2 + 96*24\tilde{L}_2^2)}{\mu^2} + \frac{48\tilde{L}_1^2}{\mu^2}\big)\eta k^{1-\alpha}C^2$$

Divide by $kE\eta/8$ on both sides to have:

$$\frac{1}{E}\sum_{e=0}^{E-1}\|\nabla h(x_{ke})\|^2 \leq \frac{8\Delta_h}{kE\eta} + 96L^2(1-\frac{k\mu\rho}{2})^{E_l}\Delta_u + 24(2\tilde{L}_1^2 + 4*96\kappa^2\tilde{L}_2^2)(1-\frac{k\mu\gamma}{2})^{E_l}\Delta_y$$

$$+ 24\big(1 + \frac{4\kappa^2(192C_f^2 + 96*24\tilde{L}_2^2)}{\mu^2} + \frac{48\tilde{L}_1^2}{\mu^2}\big)k^{-\alpha}C^2$$

By the condition $\eta < \frac{1}{4k\bar{L}}$, $\gamma < \frac{1}{128kL\kappa}$ and $\rho < \frac{1}{256kL\kappa}$, we choose $\eta = \frac{1}{8k\bar{L}}$, $\gamma = \frac{1}{256kL\kappa}$ and $\rho = \frac{1}{512kL\kappa}$, then we have:

$$\frac{1}{E}\sum_{e=0}^{E-1}\|\nabla h(x_{ke})\|^2 \leq \frac{64\bar{L}\Delta_h}{E} + 96L^2(1-\frac{k\mu\rho}{2})^{E_l}\Delta_u + 24(2\tilde{L}_1^2 + 4*96\kappa^2\tilde{L}_2^2)(1-\frac{k\mu\gamma}{2})^{E_l}\Delta_y$$

$$+ 24\big(1 + \frac{4\kappa^2(192C_f^2 + 96*24\tilde{L}_2^2)}{\mu^2} + \frac{48\tilde{L}_1^2}{\mu^2}\big)k^{-\alpha}C^2$$

Then to reach an $\epsilon$-stationary point, we need to have:

$$E \geq \frac{128\bar{L}\Delta}{\epsilon^2}, \text{ and }, k^\alpha \geq \frac{48C^2}{\epsilon^2}\bigg(1 + \frac{4\kappa^2(192C_f^2 + 96*24\tilde{L}_2^2)}{\mu^2} + \frac{48\tilde{L}_1^2}{\mu^2}\bigg)$$

and $E_l = O(log(\epsilon^{-1}))$. This completes the proof. $\square$

**Proposition B.18.** *Given Assumptions 4.3-4.4, Assumption 4.7 and Assumption 4.5 hold, then we have:* $\|\sum_{\bar{\tau}=t}^{\tau-1}\eta_{\bar{\tau}}\big(\nabla_y g(x_t, y_{x_t}) - \nabla_y g(x_{\bar{\tau}}, y_{\bar{\tau}}; \zeta_{\bar{\tau},1})\big)\|^2 \leq \eta^2 k^{2-\alpha}C^2$ *for any* $\tau \leq t + k - 1$. *The same upper bound is achieved for other properties:* $\nabla h(x)$, $\nabla_x f(x, y)$, $\nabla_y g(x, y)$, $\nabla_{y^2} g(x, y)$, $\nabla_{xy} g(x, y)$

*Proof.* This proposition can be adapted from Lemma 2 of [34]. $\square$

**Proposition B.19.** *Suppose we have function* $g(y)$, *which is L-smooth and $\mu$-strongly-convex, then suppose* $\gamma < \frac{1}{2L}$, *the progress made by one step of gradient descent is:*

$$\|y_{t+1} - y^*\|^2 \leq \big(1 - \frac{\mu\gamma}{2}\big)\|y_t - y^*\|^2 - \frac{\gamma^2}{2}\|\nabla_y g(y_t)\|^2 + \frac{4\gamma}{\mu}\|\nabla_y g(y_t) - w_t\|^2$$

*where* $y^*$ *is the minimum of* $g(y)$ *and we have update rule* $y_{t+1} = y_t - \gamma\omega_t$.

*Proof.* First, by the strong convexity of of function $g(y)$, we have:

$$g(y^*) \geq g(y_t) + \langle\nabla_y g(y_t), y^* - y_t\rangle + \frac{\mu}{2}\|y^* - y_t\|^2$$

$$= g(y_t) + \langle\nabla_y g(y_t), y^* - y_{t+1}\rangle + \langle\nabla_y g(y_t), y_{t+1} - y_t\rangle + \frac{\mu}{2}\|y^* - y_t\|^2 \qquad (37)$$

Then by $L$-smoothness, we have:

$$\frac{L}{2}\|y_{t+1} - y_t\|^2 \geq g(y_{t+1}) - g(y_t) - \langle\nabla_y g(y_t), y_{t+1} - y_t\rangle \qquad (38)$$

Combining the 37 with 38, we have:

$$g(y^*) \geq g(y_{t+1}) + \langle \nabla_y g(y_t), y^* - y_t \rangle + \gamma \langle \nabla_y g(y_t), w_t \rangle + \frac{\mu}{2}\|y^* - y_t\|^2 - \frac{L}{2}\|y_{t+1} - y_t\|^2$$

$$\geq g(y_{t+1}) + \frac{\gamma}{2}\|w_t\|^2 + \frac{\gamma}{2}\|\nabla_y g(y_t)\|^2 - \frac{\gamma}{2}\|w_t - \nabla_y g(y_t)\|^2 + \langle w_t, y^* - y_t \rangle$$

$$+ \langle \nabla_y g(y_t) - w_t, y^* - y_t \rangle + \frac{\mu}{2}\|y^* - y_t\|^2 - \frac{L\gamma^2}{2}\|\omega_t\|^2$$

$$\geq g(y_{t+1}) + \left(\frac{\gamma}{2} - \frac{L\gamma^2}{2}\right)\|w_t\|^2 + \frac{\gamma}{2}\|\nabla_y g(y_t)\|^2 - \frac{\gamma}{2}\|w_t - \nabla_y g(y_t)\|^2 + \langle w_t, y^* - y_t \rangle$$

$$+ \langle \nabla_y g(y_t) - w_t, y^* - y_t \rangle + \frac{\mu}{2}\|y^* - y_t\|^2$$

By definition of $y^*$, we have $g(y^*) \geq g(y_{t+1})$. Thus, we obtain

$$0 \geq \left(\frac{\gamma}{2} - \frac{L\gamma^2}{2}\right)\|w_t\|^2 + \frac{\gamma}{2}\|\nabla_y g(y_t)\|^2 - \frac{\gamma}{2}\|w_t - \nabla_y g(y_t)\|^2 + \langle w_t, y^* - y_t \rangle$$

$$+ \langle \nabla_y g(y_t) - w_t, y^* - y_t \rangle + \frac{\mu}{2}\|y^* - y_t\|^2 \qquad (39)$$

Considering the outer bound of the second term $\langle \nabla_y g(y_t) - w_t, y^* - y_t \rangle$, we have

$$-\langle \nabla_y g(y_t) - w_t, y^* - y_t \rangle \leq \frac{1}{\mu}\|\nabla_y g(y_t) - w_t\|^2 + \frac{\mu}{4}\|y^* - y_t\|^2$$

Combining with Eq. 39:

$$0 \geq \left(\frac{\gamma}{2} - \frac{L\gamma^2}{2}\right)\|w_t\|^2 + \frac{\gamma}{2}\|\nabla_y g(y_t)\|^2 - \left(\frac{\gamma}{2} + \frac{1}{\mu}\right)\|w_t - \nabla_y g(y_t)\|^2 + \langle w_t, y^* - y_t \rangle$$

$$+ \frac{\mu}{4}\|y^* - y_t\|^2$$

By $y_{t+1} = y_t - \gamma\omega_t$, we have:

$$\|y_{t+1} - y^*\|^2 = \|y_t - \gamma\omega_t - y^*\|^2 = \|y_t - y^*\|^2 - 2\gamma\langle \omega_t, y_t - y^* \rangle + \gamma^2\|\omega_t\|^2$$

$$\leq \left(1 - \frac{\mu\gamma}{2}\right)\|y_t - y^*\|^2 - \gamma^2\|\nabla_y g(y_t)\|^2 + L\gamma^3\|\omega_t\|^2 + \left(\gamma^2 + \frac{2\gamma}{\mu}\right)\|\nabla_y g(y_t) - w_t\|^2$$

Then since we choose $\gamma < \frac{1}{4L} < \frac{1}{\mu}$, we obtain:

$$\|y_{t+1} - y^*\|^2 \leq \left(1 - \frac{\mu\gamma}{2}\right)\|y_t - y^*\|^2 - \frac{\gamma^2}{2}\|\nabla_y g(y_t)\|^2 + \frac{4\gamma}{\mu}\|\nabla_y g(y_t) - w_t\|^2$$

This completes the proof. $\qquad \square$

**Corollary B.20.** *For Algorithm 3 and Algorithm 4, suppose Assumptions (the bounded gradient assumption is not required for the minimax problem) and conditions are satisfied as in Theorem 4.6, to reach an $\epsilon$-stationary point, we need $O(\epsilon^{-4})$ examples if they are sampled independently, while $O((\max(m,n))^p \epsilon^{-3})$ if some random-permutation-based without-replacement sampling is used.*

**Corollary B.21.** *For Algorithm 5, suppose Assumptions and conditions are satisfied as in Theorem 4.8, to reach an $\epsilon$-stationary point, we need $O(\epsilon^{-6})$ examples if they sampled independently, while $O((\max(m,n))^{2p}\epsilon^{-4})$ if examples are sampled following some random-permutation-based without-replacement sampling.*

It is straightforward to derive the above corollaries from Theorem 4.6 and Theorem 4.8, we omit the proof here.

## C Comparison of WiOR-BO with Other Acceleration Methods of Bilevel Optimization

In this section, we compare our algorithms with other variance reduction based algorithms for bilevel optimization. For the unconditional case, WiOR-BO obtain the same $O(\epsilon^{-3})$ rate as STORM-based

Table 2: **Comparisons of the Bilevel Opt. & Conditional Bilevel Opt. algorithms for finding an $\epsilon$-stationary point**. An $\epsilon$-stationary point is defined as $\|\nabla h(x)\| \leq \epsilon$. $Gc(f, \epsilon)$ and $Gc(g, \epsilon)$ denote the number of gradient evaluations *w.r.t.* $f(x, y)$ and $g(x, y)$; $JV(g, \epsilon)$ denotes the number of Jacobian-vector products; $HV(g, \epsilon)$ is the number of Hessian-vector products. $m$ and $n$ are the number of data examples for the outer and inner problems, in particular, $n$ is the maximum number of inner problems for conditional bilevel optimization. Our methods have a dependence over example numbers $(max(m, n))^q$, $q$ is a value decided by without-replacement sampling strategy and can have value in $[0, 1]$ (A herding-based permutation [33] can let $q = 0$).

| Setting | Algorithm | $Gc(f, \epsilon)$ | $Gc(g, \epsilon)$ | $JV(g, \epsilon)$ | $HV(g, \epsilon)$ |
|---|---|---|---|---|---|
| **B.O.** | BSA [17] | $O(\epsilon^{-4})$ | $O(\epsilon^{-6})$ | $O(\epsilon^{-4})$ | $O(\epsilon^{-4})$ |
| | TTSA [22] | $O(\epsilon^{-5})$ | $O(\epsilon^{-5})$ | $O(\epsilon^{-5})$ | $O(\epsilon^{-5})$ |
| | StocBiO [25] | $O(\epsilon^{-4})$ | $O(\epsilon^{-4})$ | $O(\epsilon^{-4})$ | $O(\epsilon^{-4})$ |
| | SOBA [9] | $O(\epsilon^{-4})$ | $O(\epsilon^{-4})$ | $O(\epsilon^{-r})$ | $O(\epsilon^{-4})$ |
| | MRBO [52] | $O(\epsilon^{-3})$ | $O(\epsilon^{-3})$ | $O(\epsilon^{-3})$ | $O(\epsilon^{-3})$ |
| | VRBO [52] | $O(\epsilon^{-3})$ | $O(\epsilon^{-3})$ | $O(\epsilon^{-3})$ | $O(\epsilon^{-3})$ |
| | SABA [9] | $O(max(m, n)^{2/3}\epsilon^{-2})$ | $O(max(m, n)^{2/3}\epsilon^{-2})$ | $O(max(m, n)^{2/3}\epsilon^{-2})$ | $O(max(m, n)^{2/3}\epsilon^{-2})$ |
| | SRBA [10] | $O(max(m, n)^{1/2}\epsilon^{-2})$ | $O(max(m, n)^{1/2}\epsilon^{-2})$ | $O(max(m, n)^{1/2}\epsilon^{-2})$ | $O(max(m, n)^{1/2}\epsilon^{-2})$ |
| | **WiOR-BO(Ours)** | $O((max(m, n))^q\epsilon^{-3})$ | $O((max(m, n))^q\epsilon^{-3})$ | $O((max(m, n))^q\epsilon^{-3})$ | $O((max(m, n))^q\epsilon^{-3})$ |
| **Cond. B.O.** | DL-SGD [23] | $O(\epsilon^{-4})$ | $O(\epsilon^{-6})$ | $O(\epsilon^{-4})$ | $O(\epsilon^{-4})$ |
| | RT-MLMC [23] | $O(\epsilon^{-4})$ | $O(\epsilon^{-4})$ | $O(\epsilon^{-4})$ | $O(\epsilon^{-4})$ |
| | **WiOR-CBO (Ours)** | $O((max(m, n))^q\epsilon^{-3})$ | $O((max(m, n))^{2q}\epsilon^{-4})$ | $O(n(max(m, n))^q\epsilon^{-3})$ | $O((max(m, n))^{2q}\epsilon^{-4})$ |

algorithm MRBO [52], while is slower than SVRG-type algorithms which have $O(n^q\epsilon^{-2})$ such as VRBO [52], SABA [9] and SRBA [10]. This relationship is similar to that for the single level optimization problems. However, as a SGD-type algorithm, WiOR-BO is much simpler to implement in practice compared to STORM-based and ARVG-based algorithms. More specifically, WIOR-BO has fewer hyper-parameters (learning rates for inner and outer problems) to tune compared to MRBO, which has six independent hyper-parameters, and the optimal theoretical convergence rate is achieved only if the complicated conditions among hyper-parameters are satisfied in Theorem 1 of [52] are satisfied, and this requires significant effort in practice. Next, SRBO/SABA evaluates the full hyper-gradient at the start of each outer loop, where we need to evaluate the first and second order derivatives over all samples in one step, which is very expensive for modern ML models (such as the transformer model with billions of parameters). In contrast, our WiOR-BO never evaluates the full gradient. Note that the inner loop length $I = lcm(m, n)$ is analogous to the concept of "epoch" in single level optimization: where we go over the data samples following a given order, but at each step we evaluate gradient over a mini-batch of samples. The practical advantage of our WiOR-BO is further verified by the superior performance over MRBO and VRBO in the Hyper-Data Cleaning Task.

