# OpenReview forum: "Provably Faster Algorithms for Bilevel Optimization via Without-Replacement Sampling"
_NeurIPS.cc/2024/Conference — NeurIPS 2024 poster_

### Official Review · Reviewer_H8tq · 2024-06-29

**Soundness:** 2
**Presentation:** 2
**Contribution:** 2
**Rating:** 4
**Confidence:** 3

**Summary:**

This paper proposes WiOR-(C)BO for without-replacement sampling algorithms for (conditional) bilevel optimization. The authors prove the convergence rate for both algorithms under regular assumptions, which are improved upon existing works on the complexity of $\epsilon$.

**Strengths:**

1. The assumptions are standard and the convergence rates are better than existing results.
2. The empirical behaviors are better than compared algorithms.

**Weaknesses:**

1. The improvement from order $\mathcal{O}(\epsilon^{-4})$ to $\mathcal{O}(\max\\{m,n\\}^p\epsilon^{-3})$ is not as great as stated to be $\mathcal{O}(\epsilon^{-3})$. If we can easily ignore the existence of $m$ and $n$, why not calculate the true gradients and Hessian matrices (instead of stochastic estimates) all time and reach a $\mathcal{O}(\max\\{m,n\\}\epsilon^{-2})$ rate of SOBA? Please clarify the influence of term $\\{m,n\\}^p$ so that the degree of improvement is clearer. If $p$ can be forced to 0 without additional assumptions, the result will be better.
2. The $p$ values in both $\mathcal{O}(\max\\{m,n\\}^p\epsilon^{-3})$ for bilevel optimization and $\mathcal{O}(\max\\{m,n\\}^{2p}\epsilon^{-4})$ for conditional bilevel optimization are not clear. Please clarify how $p$ is determined. By the way, the authors should have used another character since $p$ has already been used as the dimension of $x$.
3. The motivation of using without-replacement sampling is not solid. In the introduction section, the author stated that using without-replacement sampling achieves more accurate gradient estimations than with-replacement sampling. However, SOBA implemented by with-replacement sampling can achieve $\mathcal{O}(\epsilon^{-2})$ complexity which is better than $\mathcal{O}(\epsilon^{-3})$. Could you give some explanations on this?
4. The novelty in algorithmic design is limited. It looks quite like a direct combination of SOBA and natural without-replacement sampling strategies.

**Questions:**

1. The paper regards the special cases as one of the main contributions. Can the special cases be fully covered by the general cases? Are convergence results or algorithms for these cases different from what we could achieve by applying the general case?
2. How is $p$ (or $q$) in the convergence rate determined? If it can be forced to 0 by using certain sampling strategy, why not directly use that strategy? Will it contradict with other assumptions?
3. Why should we use without-replacement sampling rather than with-replacement sampling? I'm asking because I believe SOBA with moving average and with-replacement sampling has a better convergence rate of $\mathcal{O}(\epsilon^{-2})$ under regular assumptions. Please correct me if I'm wrong.
4. Can you compare the emprical performance with SOBA?

**Limitations:**

The limitations are stated as assumptions.

---

> ### Author Rebuttal · Authors · 2024-08-07
>
> Thanks for spending your valuable time reviewing our manuscript and provide insightful feedbacks. Our responses are provided as below:
>
> **Convergence Rate of SOBA**. We first want to clarify that SOBA [1] has convergence rate of $O(\epsilon^{-4})$ under the notation of our manuscript. In fact, the $\epsilon$-stationary point in our manuscript is defined as $\|\nabla h(x)\| \leq \epsilon$. Note that the SOBA paper uses an alternative definition of $\|\nabla h(x)\|^2 \leq \epsilon$.
>
> **The value of $p$**. Our WiOR-BO has convergence rate of $O(max(m,n)^p\epsilon^{-3})$, note that this rate is strictly better than the $O(\epsilon^{-4})$ rate of stocBiO, BSA and SOBA. As for the value of $p$, it relates to the specific example order we choose. More specifically, in Assumption 4.5 (i.e. how well the consecutive example gradients approximate the true gradients), we have the constant $C = \max(m,n)^pA$. For shuffle-once, we have $p=1$, for random-reshuffling, we can show $p=0.5$, while for the GraB algorithm~[2], we can reach $p=0$.  In our experiments, we consider the random-reshuffling and shuffle-once, as for the GraB algorithm, although it has favorable property that $p=0$, we need to store at least one gradient (also Hessian in the bilevel case), and this leads to $O(d^2)$ extra space, which is expensive for large scale neural networks.
>
> **Comparison with other Methods**.  In Table 1 of the manuscript, we compare WiOR-BO with methods such as stocBiO, BSA, and SOBA, which all adopt SGD-like updates. Our algorithm differs from these methods in the sampling strategy. The faster convergence rate of our algorithms demonstrates the efficacy of using without-replacement sampling. Then in Table 2 of the Appendix, we include more algorithms, some of them obtain faster convergence rate than WiOR-BO, yet this is achieved by using some form of **variance reduction technique**. For example, SABA ($O(max(m,n)^{2/3}\epsilon^{-2})$)  is based on SAGA, SRBA ($O(max(m,n)^{1/2}\epsilon^{-2})$) is based on SARAH, while MRBO ($O(\epsilon^{-3})$) is based on STORM. However our algorithm still demonstrates favorable empirical advantages over these variance-reduction enabled methods. In fact, WIOR-BO has  fewer hyper-parameters (learning rates for inner and outer problems) to tune compared to MRBO, which has six independent hyper-parameters, and the optimal theoretical convergence rate is achieved only if the complicated conditions among hyper-parameters are satisfied in Theorem 1 of the MRBO paper are satisfied, and this requires significant tuning effort in practice. Next, SRBO/SABA evaluates the full hyper-gradient at the start of each outer loop, where we need to evaluate the first and second order derivatives over all samples in one step, which is very expensive for modern ML models (such as the transformer model with billions of parameters). In contrast, **our WiOR-BO never evaluates the full gradient**. Note that the inner loop length $I = lcm(m,n)$ is analogous to the concept of "epoch" in single level optimization: where we go over the data samples following a given order, but at each step we evaluate gradient over a mini-batch of samples. The practical advantage of our WiOR-BO is further verified by the superior performance over MRBO and VRBO in the Hyper-Data Cleaning Task.
>
> **Novelty**:  Without-replacement sampling is widely used in practice in model training including bilevel optimization. However, most existing literature for bilevel optimization assumes the independent sampling assumption to simplify the analysis. To the best of our knowledge, our WiOR-BO (WiOR-CBO) is the first without-replacement sampling bilevel algorithm with theoretical guarantee.
>
> *Responses to Reviewer Questions*:
>
> *Q1: The paper regards the special cases as one of the main contributions. Can the special cases be fully covered by the general cases? Are convergence results or algorithms for these cases different from what we could achieve by applying the general case?*
>
> **R1**: Due to space limitation, the convergence results of the minimax and compositional cases are deferred to Corollary B.20 and Corollary B.21 in Appendix. For the compositional case, we can directly apply the analysis of bilevel optimization and get the same convergence rate. For the minimax case, we need to remove the bounded gradient assumption, i.e. $\|\nabla_y f(x,y, \xi) \| \leq C_f$ as this assumption conflicts with the assumption of $\mu$-strong convexity with respect to y. Once this assumption is removed, we can follow the same analysis used in bilevel optimization to establish the convergence rate for the minimax case.
>
> *Q2: Can you compare the emprical performance with SOBA?*
>
> **R2**: In the Hyper-Data Cleaning task, we have compared with SOBA in the name of WiR-BO.  We use the name WiR-BO as there are several single loop bilevel algorithms such as AmIGO [3] and FSLA [4] in the literature. We also compare with variance-reduction enable methods: MRBO and VRBO.
>
> Finally, we want to thank the reviewer once again. If anything remains unclear, please do not hesitate to let us know.
>
> References
>
> [1]. Dagréou, Mathieu, et al. "A framework for bilevel optimization that enables stochastic and global variance reduction algorithms." Advances in Neural Information Processing Systems 35 (2022): 26698-26710.
>
> [2]. Y. Lu, W. Guo, and C. M. De Sa. Grab: Finding provably better data permutations than random reshuffling. Advances in Neural Information Processing Systems, 35:8969–8981, 2022.
>
> [3]. M. Arbel and J. Mairal. Amortized implicit differentiation for stochastic bilevel optimization,2021.
>
> [4]. J. Li, B. Gu, and H. Huang. A fully single loop algorithm for bilevel optimization without hessian inverse. In Proceedings of the AAAI Conference on Artificial Intelligence, volume 36,pages 7426–7434, 2022

---

> > ### Author Response · Authors · 2024-08-13
> > **Any further concerns?**
> >
> > Dear reviewer,
> >
> > Thank you once again for spending time reviewing our manuscript. With the discussion period ending soon, we wanted to kindly check if your concerns have been fully resolved. If you have any further questions, please don't hesitate to reach out and we are very happy to discuss them.
> >
> > Best regards,
> >
> > Authors

---

> > ### Comment · Reviewer_H8tq · 2024-08-13
> >
> > Thanks for your rebuttal.
> >
> > The overall response looks good yet I'm still not satisfied with the part related to SOBA. In the rebuttal the authors say the convergence rate of SOBA is $\mathcal{O}(\epsilon^{-4})$, which is true for the stochastic case. However, in my original review, I'm talking about SOBA with full-batch true gradient, which is a deterministic case. With a single modification on SOBA (projecting variable $z^k$ to a ball of radius no greater than $C_f/\mu_g$), the convergence rate of deterministic SOBA can achieve $\mathcal{O}(\epsilon^{-2})$, thus $\mathcal{O}(\max\\{m,n\\}\epsilon^{-2})$ computation complexity in total. That's why I believe the dependence on $\max\\{m,n\\}^p$ is also of great importance and we cannot only compare the order of $\epsilon$. In view of this, when using shuffle-once where $p=1$, I do not see any advantage in the complexity bound compared with deterministic SOBA.

---

> ### Author Response · Authors · 2024-08-13
>
> Sorry for the confusion! Since in your original review, it reads "SOBA with moving average and with-replacement sampling", we interpret this as meaning the standard stochastic case (which involves sampling).
>
> Firstly, we agree with the reviewer that the value of $p$ is important and we indeed include it in Table 1 of the manuscript and also discuss them in the main text. As stated in the rebuttal, the dependence over $max(m,n)^p$ comes from the sample gradient error $||\frac{1}{k} \sum_{\tau = t}^{t+k-1} \nabla f (x_t,y_t; \xi_{\tau}^{\pi}) - \nabla f (x_t,y_t)||^2$, while the value of $p$ depends on specific sampling orders. In particular, random-reshuffling and the GraB method can force $p < 1$, which has a better dependence over $max(m,n)$ compared to deterministic algorithms. Additionally, we want to add that similar dependence over $max(m,n)^p$  exists for  single level algorithms that adopt without-replacement sampling.
>
> Next, we believe our work makes a valuable contribution to the study of bilevel optimization. Specifically, we demonstrate that the independent sampling assumption, commonly used in SGD-type bilevel algorithms, can be replaced by without-replacement sampling, leading to a reduced per-iteration cost (less number of backward passes) and an improved convergence rate (with the improvement depending on specific example orders). Furthermore, the new algorithms (WiOR-BO/WiOR-CBO) still perform SGD-type updates, making them practical for modern machine learning, in contrast to other variance-reduction-based or deterministic algorithms.

---

> > ### Comment · Reviewer_H8tq · 2024-08-14
> >
> > Thank you for the fair comment. Here's another question I'm concerned about. Lets assume we have $p<1$ for some sampling methods, then the proposed method has a better rate on $\max\\{m,n\\}$ but a worse rate on $\epsilon$ than deterministic SOBA, while the stochastic SOBA has a even better rate on $\max\\{m,n\\}$ but worse rate on $\epsilon$ than the proposed method. Thus, it seems that no matter which part dominates the convergence rate, the proposed method is not the best one in theory. Taking both $\max\\{m,n\\}$ and $\epsilon$ into considerations, it still remains unclear why the proposed method are theoretically better, and it just appears to be a trade-off choice. It would be better if the authors can theoretically justify the advantage on the complexities with any of the sampling strategies mentioned.

---

### Official Review · Reviewer_TL81 · 2024-07-10

**Soundness:** 3
**Presentation:** 3
**Contribution:** 3
**Rating:** 7
**Confidence:** 2

**Summary:**

This paper introduces new algorithms that leverage without-replacement sampling to enhance bilevel optimization. The main contributions are:
1. Introducing more practical bilevel optimization algorithms using without-replacement sampling.
2. Providing comprehensive theoretical analysis demonstrating improved convergence rates and efficiency.
Novel Algorithms: Introduces simpler, more practical bilevel optimization algorithms using without-replacement sampling.
3. Validating the effectiveness through applications in hyper-data cleaning and hyper-representation learning.

**Strengths:**

The paper's strengths can be summarized as follows:

Originality: The paper introduces a novel application of without-replacement sampling for bilevel optimization, presenting a fresh and practical approach in contrast to traditional methods.

Quality: The research exhibits high quality through rigorous theoretical analysis and comprehensive comparisons with existing methods, showcasing superior performance and efficiency.

Clarity: The paper is well-structured and clearly written, effectively communicating complex concepts and providing detailed explanations and results.

Significance: The proposed methods significantly enhance process efficiency.

**Weaknesses:**

1. Why are the upper bounds the same in Assumptions 4.3, 4.4, and 4.7?

**Questions:**

Line 71: Please use $O(\cdot)$ to denote the big O notation and $\tilde{O}(\cdot)$ to indicate that logarithmic terms are hidden.
LInes 131, 536, etc.: Some formulas exceed the line width.

---

> ### Author Rebuttal · Authors · 2024-08-07
>
> Thanks for spending your valuable time in reviewing our manuscript and provide insightful comments.
>
> Firstly, about the upper bound in Assumption 4.3, 4.4 and 4.7, we assume a common bound due to the following reasons. Note that these assumptions measure similar properties, namely the sample estimation error of first or second order derivatives, using a unified bound can simplify the analysis and makes the final convergence bound obtain simpler constant factors. In fact, we can assume different upper bounds, but the convergence rates are not affected.
>
> Next, We want to thank you for your advice and will use $\tilde{O}(\cdot)$ to indicate that logarithmic terms are hidden. Furthermore, we will adjust the equations so that they can fit into the line width.

---

> > ### Comment · Reviewer_TL81 · 2024-08-12
> >
> > Thanks to the authors' responses, I have no further questions and will keep my score.

---

> > > ### Author Response · Authors · 2024-08-12
> > >
> > > Thanks for your Reply!

---

### Official Review · Reviewer_sRYA · 2024-07-12

**Soundness:** 3
**Presentation:** 3
**Contribution:** 3
**Rating:** 7
**Confidence:** 4

**Summary:**

This paper improves the computational inefficiencies in bilevel optimization algorithms that rely on independent sampling. It proposes a novel without-replacement sampling-based algorithm, which achieves a faster convergence rate than existing methods.

**Strengths:**

This paper develops independent sampling for bilevel optimization, which save the large number of backward passes and improve the convergence rate. Experimental results show the good results compared to other bilevel baselines.

**Weaknesses:**

- The paper claimed that "Compared to independent sampling-based algorithm such as stocBiO, we require extra space to generate and store the orders of examples, but this cost is negligible compared to memory and computational cost of training large scale models." Is there any empirical result to show the memory costs?

- Projection is applied to $u_i^r$, but I found other bilevel papers (such as StocBio, BSA) don't use that. Is there any intuition or explanation about that?

- In Algorithm 2, the generation of $\\{\zeta_{\xi_t^{\pi}, t_l}^{\pi} \in \mathcal{D}_{l, \xi_t^{\pi}}. t_l\in [T_l]\\}$ is not very clear. It seems that there are $S\times n_i$ samples for inner loops given a outer sample $\xi_i$. How do you execute the permutation for the inner dataset.

- In experiments, what is the best hyper-parameters for each baselines, such inner and outer learning rates?

- What are the details of data sampling for other baselines? For other baselines, they usually need the independent training and validation set for upper and lower-level update, how did you construct data for them?

- Typos: Line 107, $w_x$ should be $u_x$.

**Questions:**

Please refer to weaknesses.

**Limitations:**

Yes.

---

> ### Author Rebuttal · Authors · 2024-08-07
>
> Thanks for spending your valuable time reviewing our manuscript and provide insightful feedbacks. Our responses are provided as below:
>
> *W1: The paper claimed that "Compared to independent sampling-based algorithm such as stocBiO, ... computational cost of training large scale models." Is there any empirical result to show the memory costs?*
>
> **R1**: In our algorithms, we maintain the permutation of example indices. For example, in the Hyper-Data Cleaning Task: we have 40000 training examples and 5000 validation examples and we store the permutations with numpy.uint16 data type and this leads to memory cost of 90KB. Meanwhile, the GPU memory related to training neural network is around 20.07MB, which is roughly 200 times greater than the memory used to store the index permutations. Note that for large scale neural networks, this ratio will be larger. Additionally, the permutation process is executed on the CPU and does not consume expensive GPU memory.
>
> *W2:Projection is applied to $u$, but I found other bilevel papers (such as StocBio, BSA) don't use that. Is there any intuition or explanation about that?*
>
> **R2**: Note that our algorithms perform single loop update, i.e. update $x_t$, $y_t$ and $u_t$ alternatively; in contrast, stocBiO and BSA are double loop algorithms, where for each $x_t$, they use a loop to estimate $y_{x_t}$ and the corresponding hyper-gradient, and there is no variable $u_t$ involved. Recall that  $u_t$ is used to estimate the solution of the quadratic problem in hyper-gradient computation, i.e. $u_x =  \nabla_{y^2} g(x, y_x)^{-1} \nabla_y f(x, y_x)$ and we only apply projection operation to $u_t$, therefore, stocBiO and BSA does not need the projection. In fact, this projection operation guarantee $u_t$ is always bounded during the update and is essential in the analysis. The projection operation can be found in recent single loop bilevel algorithms such as SRBA in [1].
>
> *W3: In Algorithm 2, the generation of is not very clear. It seems that there are  samples for inner loops given a outer sample. How do you execute the permutation for the inner dataset.*
>
> **R3**: As we state in line 149-150 of main text, for the inner dataset, we concatenate $S$ random permutations of the inner dataset $D_{l,\xi_t^{\pi}}$ to get {$\zeta_{\xi_{t}^{\pi}, t_l}^{\pi} \in D_{l,\xi_t^{\pi}}, t_l \in [T_l]$} with a sequence length $T_l = S\times n_{\xi_i}$. To put it simply, for each given outer example $\xi_{t}^{\pi}$, we go through its corresponding inner dataset $S$ rounds, and for each round we go through the inner dataset following a random permutation.  Finally, we correct a typo in the Line 7 of Algorithm 2: it should be {$\zeta_{\xi_i, j}^{\pi}, j\in[n_{i}]$} instead of {$\zeta_{\xi_i, j}^{\pi}, i\in[n_{i}]$}.
>
> *W4:In experiments, what is the best hyper-parameters for each baselines, such inner and outer learning rates?*
>
> **R4**: For the invariant risk minimization task: the inner and outer learning rates are 0.001; for the hyper-data cleaning task: reverse, stocBiO, BSA, AID-CG, WiR-BO, WiOR-BO-SO, WiOR-BO-RR use inner learning rate 0.1 and outer learning rate 1000, WiR-BO, WiOR-BO-SO, WiOR-BO-RR set $\rho = 0.1$; MRBO uses $d=10$, $m=500$, $c_1=0.9$, $c_2=0.9$, $\gamma=500$, $\lambda=0.2$; VRBO uses $q=5$, $\alpha=1000$, $\beta=0.2$; for the hyper-representation learning task: for the Omniglot dataset, the outer learning rate is 0.1, the inner learning rate is 0.4, $\rho = 0.002$; for the MiniImageNet dataset, the outer learning rate is 0.05, the inner learning rate is 0.01, $\rho = 0.002$, for the RL-MLMC baseline, we set $K_{max} = 6$.
>
> *W5: What are the details of data sampling for other baselines? For other baselines, they usually need the independent training and validation set for upper and lower-level update, how did you construct data for them?*
>
> **R5**: In our empirical implementation, we use a dataloader class to handle example sampling. For our algorithms, the dataloader returns examples following the order of a permutation, while for other baselines that require independent sampling, the dataloader will randomly sample a mini-batch of samples (sample indices) from the whole dataset.
>
> *W6: Typos: Line 107 $w_x$, should be $u_x$.*
>
> **R6**: We will correct this typo and carefully proofreading our manuscript.
>
> References
>
> [1]. Dagréou, Mathieu, et al. "A lower bound and a near-optimal algorithm for bilevel empirical risk minimization." International Conference on Artificial Intelligence and Statistics. PMLR, 2024.

---

> > ### Comment · Reviewer_sRYA · 2024-08-11
> >
> > Thanks for your rebuttal, I would like to keep my rating.

---

> > > ### Author Response · Authors · 2024-08-12
> > >
> > > Thanks for your reply!

---

### Official Review · Reviewer_Qw9k · 2024-07-13

**Soundness:** 3
**Presentation:** 2
**Contribution:** 3
**Rating:** 6
**Confidence:** 3

**Summary:**

This paper investigates stochastic bilevel optimization. One common practice herein is that the data samples used to calculate all the derivatives are mutually independent, which may however introduce more computational cost compared to the case where the samples are reused. Motivated by this, the authors explore stochastic bilevel optimization without using independent sampling, particularly under the without-replacement sampling scheme. Based on previous single-loop algorithm, two with-replacement bilevel optimization algorithms are proposed for standard bilevel optimization and conditional bilevel optimization, respectively, which can also be applied to several special cases. The authors show that faster convergence rates can be achieved by the proposed algorithms compared to the counterparts with independent sampling. Experiments on multiple tasks are conducted to further justify the performance of the proposed algorithms.

**Strengths:**

1. The idea of investigating bilevel optimization without independent sampling is interesting and novel.

2. Algorithms and theoretical results are provided for various setups with a faster convergence rate.

3. The experimental results also clearly show the superior performance of the proposed algorithms.

**Weaknesses:**

1. It is not clear what the key technical contributions in the theoretical analysis are.

2. As the authors claim the extension to the special cases of minimax and compositional optimization problems as a contribution, it is better to compare with the baselines in these problems.

3. The presentation can be further improved. For examples, the figures in the experiments are not clear because of the small font size.

**Questions:**

1. How was the bias due to the without-replacement sampling handled in the analysis? Using window averaging? In this case, did you need to store the gradients of previous samples?

2. When you compared the convergence rate of algorithm 1 with baselines, e.g., stocBiO, how did you justify that the faster convergence is because of the sampling but not the single-loop update? Given that stocBiO is not a single-loop algorithm.

**Limitations:**

Yes

---

> ### Author Rebuttal · Authors · 2024-08-07
>
> Thanks for spending your valuable time reviewing our manuscript and provide insightful feedbacks. Our responses are provided as below:
>
> **Technical contributions**. We provide the first theoretical analysis of bilevel optimization that uses without-replacement sampling. A key insight in our analysis is to *analyze the descent of potential function over an interval of length k instead of one step*. This is important as only the amortized gradient estimation error ($|\frac{1}{k}\sum_{\tau=t}^{t+k-1}\nabla f(x,y,\xi_\tau) - \nabla f(x,y)|$) of without-sampling exhibits improvement compared to independent sampling. Meanwhile, we need to carefully balance the intertwined hyper-gradient estimation, inner variable estimation error and the estimation error to $u_x$ in the analysis.
>
> **Fair-Classification Task**: we provide further experimental results over a fair-classification task [1,2], which can be formulated as a minimax optimization problem. More specifically, for a classification task with $K$ categories, the objective minimizes the maximum loss over all categories, this improves the fairness among different categories. The objective is as follows:
>
> $$ \min_{w}\max_{u \in U} \sum_{i=1}^K u_i L_i(w) - \lambda ||u - \frac{1}{K}||^2$$
>
> where $u_i$ is the weight for category $i$ and $U$ denotes the probability simplex, $L_i(w)$ denotes the classification loss over category $i$ and $w$ is the model parameter. Follow the setting in [2], we consider  T-shirt/top, Coat and Shirt categories
> in the Fashion-MNIST dataset (K = 3) and solve the objective with SGDA [3], WiOR-MiniMax-SO and WiOR-MiniMax-RR. Note that SGDA performs the minimization and maximization alternatively and differs our WiOR-MiniMax-SO(-RR) only in the sampling strategy. Be We show results in the table below, we show the training loss across different epochs:
>
> | Method \ Epochs             | 1     | 5     | 10    | 20    | 30    | 50    |
> |---------------------|-------|-------|-------|-------|-------|-------|
> | SGDA                | 0.703 | 0.339 | 0.266 | 0.223 | 0.176 | 0.164 |
> | WiOR-MiniMax-SO     | 0.654 | 0.319 | 0.252 | 0.207 | 0.162 | 0.125 |
> | WiOR-MiniMax-RR     | 0.648 | 0.322 | 0.248 | 0.208 | 0.160 | 0.133 |
>
> As shown in the table, our algorithms outperforms SGDA, which demonstrate the efficacy of using without-replacement sampling.
>
> *Responses to Reviewer Questions*
>
> *Q1: How was the bias due to the without-replacement sampling handled in the analysis? Using window averaging? In this case, did you need to store the gradients of previous samples?*
>
> **A1**: We do not need to store the gradients of previous samples, and the window averaging is solely for analysis purpose. For example to bound the estimation error of $\nabla_x f(x_t, y_t; \xi_{t})$ to $\nabla_x f(x_t, y_t)$, we bound the following term $||\sum_{\tau=t}^{t+k} (\nabla_x f(x_\tau, y_\tau; \xi_{\tau}) - \nabla_x f(x_t, y_t))||^2$ and this term can be further broken into two sub-terms: $||\sum_{\tau=t}^{t+k} (\nabla_x f(x_t, y_t; \xi_{\tau}) - \nabla_x f(x_t, y_t))||^2$ and $||\sum_{\tau=t}^{t+k} (\nabla_x f(x_\tau, y_\tau; \xi_{\tau}) - \nabla_x f(x_t, y_t; \xi_{\tau}))||^2$. For the first term, the error decrease as $k$ increase (as we will go through more samples due to without replacement sampling), while for the second term, it is equivalent to bound $||(x_\tau, y_\tau) - (x_t, y_t)||^2$ by the smoothness assumption and this variable drift term can be bounded by tuning the value of learning rates.
>
> *Q2: When you compared the convergence rate of algorithm 1 with baselines, e.g., stocBiO, how did you justify that the faster convergence is because of the sampling but not the single-loop update? Given that stocBiO is not a single-loop algorithm.*
>
> **A2**: The rate improvement is indeed due to the without-replacement sampling strategy. SOBA[4] also adopts a single loop update rule, but this algorithm also obtains the same $O(\epsilon^{-4})$ rate as stocBiO.
>
> Finally, thanks for your review again. We will also carefully polish our presentation, including the font sizes in the figures suggested in your review.
>
> References:
>
> [1] Nouiehed, Maher, et al. "Solving a class of non-convex min-max games using iterative first order methods." Advances in Neural Information Processing Systems 32 (2019).
>
> [2]. Huang, Feihu, Xidong Wu, and Zhengmian Hu. "Adagda: Faster adaptive gradient descent ascent methods for minimax optimization." International Conference on Artificial Intelligence and Statistics. PMLR, 2023.
>
> [3]. Lin, T., Jin, C., and Jordan, M. (2020a). On gradient descent ascent for nonconvex-concave minimax problems. In International Conference on Machine Learning, pages 6083–6093. PMLR.
>
> [4]. M. Dagréou, P. Ablin, S. Vaiter, and T. Moreau. A framework for bilevel optimization that enables stochastic and global variance reduction algorithms. arXiv preprint arXiv:2201.13409,2022.

---

> > ### Author Response · Authors · 2024-08-13
> > **Additional Experiments**
> >
> > Below, we show experimental results for one more task:  Risk-averse portfolio management. In this task, we are given a set of assets with their returns at different time slots and then we need to design a portfolio with the objective of maximum mean return and minimum variance across time slots. This task can be formulated as a *compositional optimization problem* (See [1] for a more detailed description). We compare our WiOR-Comp-RR and WiOR-Comp-SO with the SCGD [2], for our methods, we set $\gamma=0.99$, $\rho=0.99$ and $\eta=0.001$, and for SCGD, we set $\alpha=0.001$ and $\beta=0.99$. Table below shows the objective loss over number of iterations (we set batch-size as 100):
> >
> > |  Iters     | 1     | 500   | 1000  | 1500  | 2000  |
> > |-------|-------|-------|-------|-------|-------|
> > | SCGD  | 0.985 | 0.379 | 0.137 | 0.057 | 0.029 |
> > | WiOR-Comp-SO | 0.982 | 0.335 | 0.122 | 0.041 | 0.015 |
> > | WiOR-Comp-RR | 0.913 | 0.331 | 0.118 | 0.037 | 0.014 |
> >
> > As shown by the experimental results, our algorithms outperforms the baseline SCGD.
> >
> > Finally, we would like to once again thank the reviewer for their time and effort in reviewing our manuscript. Please do not hesitate to reach out if you have any further questions.
> >
> > Reference
> >
> > [1]. Chen, Tianyi, Yuejiao Sun, and Wotao Yin. "Solving stochastic compositional optimization is nearly as easy as solving stochastic optimization." IEEE Transactions on Signal Processing 69 (2021): 4937-4948.
> >
> > [2]. Wang, Mengdi, Ethan X. Fang, and Han Liu. "Stochastic compositional gradient descent: algorithms for minimizing compositions of expected-value functions." Mathematical Programming 161 (2017): 419-449.

---

### Official Review · Reviewer_1Xqw · 2024-07-30

**Soundness:** 3
**Presentation:** 3
**Contribution:** 3
**Rating:** 4
**Confidence:** 4

**Summary:**

The paper introduces a new algorithm called WiOR-BO, which leverages without-replacement sampling to achieve faster convergence rates compared to traditional independent sampling methods. The algorithm is applied to standard bilevel optimization, conditional bilevel optimization, and specific cases such as minimax and compositional optimization problems. The authors provide a theoretical analysis demonstrating that WiOR-BO achieves a significantly improved convergence rate over existing methods. Empirical validation on synthetic and real-world tasks, including hyper-data cleaning and hyper-representation learning, shows the superior performance and computational efficiency of the proposed algorithms.

**Strengths:**

By employing without-replacement sampling, the algorithm effectively reduces the number of necessary backward passes, leading to substantial computational savings. This is especially beneficial in large-scale machine learning models where back-propagation can be expensive. The algorithm's design allows for simultaneous updates of upper and lower-level parameters, improving convergence rates and computational efficiency compared to traditional methods. Additionally, the theoretical guarantees provided on convergence offer a robust foundation for its application, making the algorithm a valuable tool for tasks such as hyperparameter optimization and data cleaning. Its ability to utilize a finite subset of data without replacement ensures reduced variance in gradient estimates, enhancing the stability and reliability of the optimization process.

**Weaknesses:**

The authors have selected learning rates and other tuning parameters to optimize performance, but they do not report any sensitivity analysis regarding these choices. This lack of analysis leaves open questions about the robustness of the algorithm's performance under different parameter settings. Additionally, there is no provided methodology or algorithm for efficiently selecting these parameters without the risk of overfitting to the validation set, which could undermine the generalizability of the results.

Furthermore, the proposed algorithms are designed under the assumption of a static dataset, which restricts their applicability in scenarios where data arrives sequentially, such as in dynamic or online learning environments. This limitation could be mitigated by extending the algorithms to handle streaming data or datasets that evolve over time, making the methods more versatile and applicable to real-world situations where data distribution may change. Addressing these issues would enhance the robustness and broader applicability of the proposed solutions.

**Questions:**

1. Comparative Benchmarks: The paper provides empirical validation, but could the authors include more comprehensive comparisons with state-of-the-art methods, especially in dynamic environments? How does the proposed method perform in cases where data distributions shift over time?
2. Computational Efficiency: While the method reduces the number of backward passes, how does it scale with very large datasets or high-dimensional data?
3. Hyperparameter Tuning: Could the authors elaborate on the process of selecting hyperparameters, such as learning rates and other tuning parameters? Specifically, are there any strategies employed to avoid overfitting during hyperparameter tuning? Additionally, how sensitive is the performance of the algorithm to these hyperparameter choices, and have the authors conducted any sensitivity analyses to assess this? Addressing these points could provide more clarity on the robustness and practical implementation of the proposed algorithms.

**Limitations:**

Are there any specific limitations that the authors anticipate in applying this method to different domains or tasks? What future directions do the authors envision for improving the algorithm, particularly concerning hyperparameter tunning, adaptivity and scalability?

---

> ### Author Rebuttal · Authors · 2024-08-07
>
> Thanks for spending your valuable time reviewing our manuscript and provide insightful feedbacks. We provide our responses to your comments below:
>
> **Hyper-parameter Tuning**. Our algorithms include three hyper-parameters: the outer learning rate $\eta$, the inner learning rate $\gamma$ and the learning rate of variable $u$: $\rho$, and they are easy to tune and robust to specific choices of hyper-parameters. More specifically, we use **grid search** (as stated in the manuscript) to perform hyper-parameter tuning, which is a standard practice used by optimization literature [1,2], furthermore, we use the loss over a hold-out validation set as the metric for tuning and adopt the early stoping technique. Take the hyper-data cleaning task as an example, we search $\eta$ in {1, 10, 100, 1000, 10000}, $\gamma$ in {0.001, 0.01, 0.1, 0.5, 1} and $\rho$ in {0.001, 0.01, 0.02, 0.05, 0.1}. Similar to the behavior of the learning rate in SGD for single-level problems, the performance initially improves as we increase the learning rate, and it will start to diverge beyond a certain point.  In the following experiments, we perform ablation studies to the learning rates for our WiOR-BO algorithm over the hyper-data cleaning task, where show the validation loss vs Number of Hyper-iterations:
>
> **Table 1**: Ablation Study over $\gamma$  ($\eta = 1000$ and $\rho = 0.1$)
>
> | $\gamma$ \ Iters  | 1    | 500  | 1000 | 2000 |
> |---|------|------|------|------|
> | 0.05 | 3.543 | 0.555 | 0.303 | 0.256 |
> | 0.1  | 3.154 | 0.501 | 0.239 | 0.157 |
> | 0.2  | 2.948 | 0.511 | 0.253 | 0.177 |
>
> **Table 2**: Ablation Study over $\eta$  ($\gamma = 0.1$ and $\rho = 0.1$)
>
> | $\eta$ \ Iters  | 1    | 500  | 1000 | 2000 |
> |---|------|------|------|------|
> | 100 | 3.238 | 0.656 | 0.465 | 0.256 |
> | 1000  | 3.154 | 0.501 | 0.239 | 0.157 |
> | 10000 | 3.014 | 0.544 | 0.335 | 0.208 |
>
> **Table 3**: Ablation Study over $\rho$  ($\gamma = 0.1$ and $\eta = 1000$)
>
> | $\rho$ \ Iters  | 1    | 500  | 1000 | 2000 |
> |---|------|------|------|------|
> | 0.05 | 3.239 | 0.571 | 0.329 | 0.213 |
> | 0.1  | 3.154 | 0.501 | 0.239 | 0.157 |
> | 0.2 | 2.961 | 0.518 | 0.378 | 0.228 |
>
> As shown by the results in Tables 1 to 3, the best performance is achieved with $\eta=1000$, $\gamma=0.1$, and $\rho=0.1$. The performance with other hyperparameter settings is also comparable.
>
> **Computational Efficiency**. We can discuss the influence of dataset size and model size based on the convergence rate. As stated in Table-1 of the manuscript, WiOR-BO has convergence rate of $max(m,n)^q\epsilon^{-3}$ and WiOR-CBO has rate of $max(m,n)^{2q}\epsilon^{-4}$. Note that the convergence rate is measured by the number of gradient queries and our convergence results show that this number is not affected by the model size, however, the cost of one gradient query indeed increases as the model size increase; as for the dataset size, the convergence rate of our model has a dependence of $max(m,n)^q$, where $q$ is a constant that relates to the without-replacement sampling strategy we use. For shuffle-once, we have $q=1$; for random-reshuffling, we have $q=0.5$, and if use the GraB algorithm in [3] (which is based on a herding and balancing), $q=0$.
>
> **Dynamic Environments**. In our manuscript, we focus on the finite-sum setting (i.e. over a static dataset), which is a standard setting in modern machine learning [4], and we also believe it is a suitable setting to study the effect of example correlation effect (due to without-replacement sampling) to the convergence of bilevel optimization. While the dynamic environment setting is an interesting extension, it is beyond the scope of this manuscript and warrants a separate, dedicated study.
>
> Finally, in terms of the limitation, since the convergence results are based on various regularity assumptions to the outer and inner problems, real world tasks might not hold these assumptions. As for the future direction,  we can study the effect of without-replacement sampling under other assumptions, in this manuscript, we consider the nonconvex-strongly-convex class of functions, and we could extend our analysis to other classes of functions, furthermore, the reviewer's advice to consider dynamic environments is also possible.
>
> References:
>
> [1]. Yang, Junjie, Kaiyi Ji, and Yingbin Liang. "Provably faster algorithms for bilevel optimization." Advances in Neural Information Processing Systems 34 (2021): 13670-13682.
>
> [2]. Ji, Kaiyi, Junjie Yang, and Yingbin Liang. "Bilevel optimization: Convergence analysis and enhanced design." International conference on machine learning. PMLR, 2021.
>
> [3]. Lu, Yucheng, Wentao Guo, and Christopher M. De Sa. "Grab: Finding provably better data permutations than random reshuffling." Advances in Neural Information Processing Systems 35 (2022): 8969-8981.
>
> [4]. Johnson, Rie, and Tong Zhang. "Accelerating stochastic gradient descent using predictive variance reduction." Advances in neural information processing systems 26 (2013).

---

> > ### Comment · Reviewer_1Xqw · 2024-08-12
> >
> > Thank you for your replies. This addresses some of the points I raised and I decide to maintain my score.

---

> > > ### Author Response · Authors · 2024-08-13
> > >
> > > Thanks for your reply. Could you please elaborate any concerns that may still be unclear? We would be very happy to discuss them further.

---

### Decision · Program_Chairs · 2024-09-25

**Decision:**

Accept (poster)

**Comment:**

The paper introduces WiOR-BO, a new algorithm that leverages without-replacement sampling to enhance the efficiency of bilevel optimization. The method is designed to address the limitations of traditional SGD-type bilevel algorithms, which rely on independent sampling. WiOR-BO is applicable to standard bilevel optimization, conditional bilevel optimization, and specific cases such as minimax and compositional optimization problems. The authors provide theoretical convergence guarantees and demonstrate the algorithm's effectiveness through empirical validation on multiple tasks.

Pros:

Novel Application: Introduces without-replacement sampling to bilevel optimization, a relatively unexplored area that offers potential benefits in terms of computational efficiency.
Theoretical Contribution: Provides a comprehensive theoretical analysis, demonstrating improved convergence rates compared to existing methods.
Empirical Validation: Strong empirical results on various tasks, showing the effectiveness of the proposed algorithms.
Practical Relevance: The proposed method offers a practical alternative to existing bilevel optimization algorithms, particularly in large-scale machine learning scenarios.

Cons:

Limited Novelty in Algorithmic Design: The method is perceived as a direct combination of existing techniques (e.g., SOBA) with without-replacement sampling, limiting the novelty.
Unclear Theoretical Advantage: Despite the theoretical contributions, the actual advantage of the proposed method over deterministic and stochastic SOBA remains unclear, especially when considering both gradient and Hessian terms.
Complexity Considerations: The improvement in convergence rates is not as significant as initially claimed, leading to questions about the overall theoretical superiority of the method.
Presentation Issues: The clarity of some figures and explanations in the paper could be improved, particularly in terms of technical details and parameter settings.

Thins that should be addressed when preparing the final version of the paper.

-  There are lingering concerns about the theoretical advantage of the proposed method over existing methods like deterministic and stochastic SOBA, particularly in terms of the trade-off between gradient and Hessian complexities.

-  While the empirical results are strong, the theoretical justification for why the proposed method is preferable in all scenarios remains insufficiently addressed.

The paper offers a meaningful contribution to bilevel optimization, particularly through its introduction of without-replacement sampling. However, the novelty and theoretical justification of the method require further clarification. The empirical results are compelling, but the paper would benefit from additional theoretical insights and clearer presentation of its advantages over existing methods. These issues can be addressed when preparing a camera ready version.